# Experience shapes chandelier cell function and structure in the visual cortex

Koen Seignette[1], Nora Jamann[2,3], Paolo Papale[4], Huub Terra[1], Ralph O Porneso[1], Leander de Kraker[1], Chris van der Togt[1,4], Maaike van der Aa[1], Paul Neering[1,4], Emma Ruimschotel[1], Pieter R Roelfsema[4,5,6,7], Jorrit S Montijn[8], Matthew W Self[4], Maarten HP Kole[2,3], Christiaan N Levelt[1,9]*

[1]Department of Molecular Visual Plasticity, Netherlands Institute for Neuroscience, Amsterdam, Netherlands; [2]Department of Axonal Signaling, Netherlands Institute for Neuroscience, Amsterdam, Netherlands; [3]Department of Biology Cell Biology, Neurobiology and Biophysics, Faculty of Science, Utrecht University, Utrecht, Netherlands; [4]Department of Vision & Cognition, Netherlands Institute for Neuroscience, Amsterdam, Netherlands; [5]Laboratory of Visual Brain Therapy, Sorbonne Université, Institut National de la Santé et de la Recherche Médicale, Centre National de la Recherche Scientifique, Institut de la Vision, Paris, France; [6]Department of Integrative Neurophysiology, Centre for Neurogenomics and Cognitive Research, VU University, Amsterdam, Netherlands; [7]Department of Psychiatry, Academic Medical Center, University of Amsterdam, Amsterdam, Netherlands; [8]Department of Cortical Structure & Function, Netherlands Institute for Neuroscience, Amsterdam, Netherlands; [9]Department of Molecular and Cellular Neurobiology, Center for Neurogenomics and Cognitive Research, VU University Amsterdam, Amsterdam, Netherlands

*For correspondence:
c.levelt@nin.knaw.nl

**Competing interest:** The authors declare that no competing interests exist.

**Abstract** Detailed characterization of interneuron types in primary visual cortex (V1) has greatly contributed to understanding visual perception, yet the role of chandelier cells (ChCs) in visual processing remains poorly characterized. Using viral tracing we found that V1 ChCs predominantly receive monosynaptic input from local layer 5 pyramidal cells and higher-order cortical regions. Two-photon calcium imaging and convolutional neural network modeling revealed that ChCs are visually responsive but weakly selective for stimulus content. In mice running in a virtual tunnel, ChCs respond strongly to events known to elicit arousal, including locomotion and visuomotor mismatch. Repeated exposure of the mice to the virtual tunnel was accompanied by reduced visual responses of ChCs and structural plasticity of ChC boutons and axon initial segment length. Finally, ChCs only weakly inhibited pyramidal cells. These findings suggest that ChCs provide an arousal-related signal to layer 2/3 pyramidal cells that may modulate their activity and/or gate plasticity of their axon initial segments during behaviorally relevant events.

## eLife assessment

This **important** work shows **compelling** evidence that Chandelier cells in the visual cortex receive inputs most prominently from local layer 5 pyramidal neurons, only mildly inhibit L2/3 pyramidal neurons, and respond massively to visuomotor mismatch. It also indicates that visual experience in the virtual tunnel activates a plasticity mechanism in Chandelier cells which could be due to the particular visuo-motor coupling experienced in this setting, although a specific control is lacking for this conclusion. This study will be of interest to neuroscientists involved in cortical circuits, visual processing, and predictive coding research.

## Introduction

The neocortex contains a diverse set of inhibitory interneuron types. The characterization of their connectivity and functions has greatly contributed to our comprehension of cortical circuits and their role in visual perception. The realization that disinhibitory circuits can regulate visual responses based on context has helped to understand mechanisms underlying attention, visual segmentation, predictive processing, and plasticity (*Karnani et al., 2016*; *Kirchberger et al., 2023*; *Kirchberger et al., 2021*; *Pfeffer et al., 2013*; *van Versendaal and Levelt, 2016*; *Zhang et al., 2014*; *Attinger et al., 2017*; *Adesnik et al., 2012*; *Keller et al., 2020*; *Fu et al., 2014*). While most cortical interneuron subsets in primary visual cortex (V1) are well characterized, much remains unknown about the axo-axonic chandelier cells (ChCs) (*Peters et al., 1982*; *Jones, 1975*) due to the difficulty of genetically targeting them. The recent discovery that vasoactive intestinal peptide receptor 2 (Vipr2) is a marker for cortical ChCs and the availability of Vipr2-Cre mice have now made it possible to perform thorough analyses of this enigmatic cell type (*Schneider-Mizell et al., 2021*; *Tasic et al., 2018*).

ChCs are unique among interneuron types in that they exclusively innervate pyramidal cells (PyCs) at their axon initial segment (AIS), the site where action potentials (APs) are generated (*Somogyi, 1977*; *Kole and Stuart, 2012*). This anatomical organization has led to the idea that ChCs may exert powerful control over AP generation (*Veres et al., 2014*). However, there is considerable controversy about whether ChC innervation of the AIS causes inhibition or excitation of PyCs (*Veres et al., 2014*; *Pan-Vazquez et al., 2020*; *Szabadics et al., 2006*; *Woodruff et al., 2009*; *Woodruff et al., 2011*; *Woodruff et al., 2010*; *Glickfeld et al., 2009*; *Molnár et al., 2008*; *Murata and Colonnese, 2020*; *Lipkin and Bender, 2023*). A recent study found that ChCs in primary somatosensory cortex (S1) depolarize the AIS during the first 2–3 weeks after birth, while they cause hyperpolarization or shunting in adult mice (*Pan-Vazquez et al., 2020*). Accordingly, the few studies that have manipulated ChC activity in adult mice in vivo also found an inhibitory effect on PyCs (*Lu et al., 2017*; *Dudok et al., 2021*).

On top of the limited understanding of their impact on neuronal excitability, there is also little known about the connectivity of ChCs in V1 and their response properties (*Jung et al., 2022*). In prelimbic cortex, ChCs receive input from local and contralateral PyCs in deep layer 3, while they preferentially innervate more superficial amygdala-innervating PyCs (*Lu et al., 2017*). This non-reciprocal connectivity pattern suggests that ChCs may establish a hierarchical relationship among cortical networks. Interestingly, superficial layer 2/3 (L2/3) PyCs in V1 also receive more ChC synapses than deep layer 3 PyCs (*Schneider-Mizell et al., 2021*), but whether connections are non-reciprocal remains unknown. Recent studies using in vivo two-photon calcium imaging in V1 show that ChC activity is highly correlated with pupil size and locomotion (*Schneider-Mizell et al., 2021*; *Bugeon et al., 2022*), indicating arousal-related ChC activity consistent with what has been observed in other brain regions (*Dudok et al., 2021*; *Bienvenu et al., 2012*; *Massi et al., 2012*). However, visually evoked activity has also been observed in V1 ChCs (*Bugeon et al., 2022*). This activity profile is similar to that of vasoactive intestinal peptide (VIP)+ interneurons and neurogliaform cells (*Bugeon et al., 2022*), which are both known to receive strong top-down inputs from higher-order cortical areas. It is not known whether this is also true for ChCs, but if so, an interesting possibility would be that ChCs may establish a hierarchical relationship among cortical networks.

ChCs have also been implicated in regulating various forms of plasticity. In adult S1, it was found that ChCs increase the number of synapses at the AIS if their postsynaptic targets are chemogenetically activated, suggesting that ChCs may play a role in homeostatic control of PyC activity (*Pan-Vazquez et al., 2020*). A contribution to homeostatic scaling of neuronal output is consistent with the observation that the size of the PyC soma is proportional to the number of ChC synaptic contacts (*Schneider-Mizell et al., 2021*; *Veres et al., 2014*). Furthermore, in CA1 it was discovered that optogenetic suppression of ChCs during spatial exploration favors place field remapping (*Dudok et al., 2021*). In premotor cortex it was found that suppressing the influence of ChCs reduced performance in a learned motor task (*Jung et al., 2023*). Finally, in the binocular zone of developing V1, elimination of ChCs during development was found to be crucial for the maturation of inputs from the ipsilateral eye and depth perception (*Wang et al., 2021*). Together, these studies suggest that ChCs may regulate plasticity by directly altering the excitability of their targets at the AIS or reducing the ability of PyCs to undergo changes of their excitatory synaptic inputs.

Here, we analyzed ChCs in L2/3 of V1 to understand their role in visual processing and plasticity. We find that ChCs receive inputs from local L5 PyCs and higher cortical regions and exhibit weak selectivity for visual stimulus content. Imaging ChC activity in mice running through a virtual tunnel showed that they respond to events that are known to increase arousal levels, such as locomotion and visuomotor mismatch. Surprisingly, visuomotor experience in the virtual tunnel strongly decreased ChC visual responses. It also resulted in plasticity of the length of PyC AISs and their innervation by ChCs. Finally, ChCs exerted only mild inhibitory influence on PyCs, affecting only a small proportion of cells. Our findings suggest that ChCs predominantly respond to arousal related to locomotion or unexpected events/stimuli, and act to weakly modulate activity and/or gate plasticity of L2/3 PyCs in V1.

## Results

### ChCs receive input from long-range sources and L5 PyCs in V1

We first identified the sources of synaptic input to ChCs in V1 using trans-synaptic retrograde rabies tracing. In order to label ChCs in layer 2/3, we made use of Vipr2-Cre mice, in which Cre recombinase is selectively expressed in cortical ChCs (*Schneider-Mizell et al., 2021*; *Tasic et al., 2018*; *Daigle et al., 2018*). In superficial V1 of these mice, we injected Cre-dependent AAV vectors expressing the avian glycoprotein EnvA receptor TVA, rabies glycoprotein (G) and eYFP on day 1, followed by a glycoprotein-deleted (dG) rabies virus on day 27 (*Lee and Kim, 2019*; *Figure 1A, B*). Most labeled neurons were located on the border between L1 and L2/3 and displayed typical ChC morphology (*Figure 1—figure supplement 1A, B*). Quantification based on morphological properties (eYFP+ cell bodies at the L1/L2 border and L1 dendrites) revealed that 87% (287/329) of labeled neurons were ChCs (*Figure 1—figure supplement 1A, B*). This specificity matches that of Cre driver lines for other inhibitory types (*Taniguchi et al., 2011*). We tested the specificity of the rabies virus by injecting it without the AAV helper vectors and found no labeled neurons (*Figure 1—figure supplement 1C*). We then quantified neurons providing monosynaptic input to ChCs across the brain of four mice also injected with the AAV helper vectors. This revealed that ChCs received long-range inputs from various thalamic and cortical regions (e.g. dorsal lateral geniculate nucleus, lateral posterior nucleus, retrosplenial cortex, and S1), matching long-range inputs described for other interneuron subsets in V1 (*Ma et al., 2021*). The most abundant sources of presynaptic partners of ChCs, however, were found locally in L5 and to a lesser extent in L1–4 of V1 (*Figure 1C*). Labeled L5 neurons had pyramidal-shaped cell bodies and dendritic spines, indicating that L5 inputs to ChCs are excitatory (*Figure 1—figure supplement 1D–F*). This local innervation pattern is reminiscent of ChCs in S1 (*Xu and Callaway, 2009*), but differs significantly from the innervation pattern of ChCs in the prefrontal cortex where they predominantly receive input from contralaterally projecting PyCs in deep layer 3 (*Lu et al., 2017*).

To test the monosynaptic nature of the long-range input cells observed with rabies tracing, we used optogenetic stimulation in combination with electrophysiological recordings in acute V1 slices. For these experiments we injected an AAV vector driving expression of ChR2-eYFP in retrosplenial cortex for optogenetic stimulation. This area was chosen because it contained more input neurons than any other brain area that was sufficiently distal from V1 to prevent potential leakage of the viral vector into V1 itself. We also injected a Cre-dependent AAV vector driving mCyRFP1 expression in V1 to label ChCs and performed whole-cell recordings in V1 slices 3 weeks later (*Figure 1D–F*). Local optogenetic activation of RSC boutons in V1 generated inward currents in ~85% (11/13) of voltage-clamped ChCs. The resulting excitatory postsynaptic potentials (PSPs) were abolished in the presence of tetrodotoxin (TTX), but reappeared upon additional application of the potassium-channel blocker 4-aminopyridine (4-AP) (*Figure 1H*), which facilitates optogenetically evoked synaptic release in absence of AP generation. These data corroborate the idea that RSC inputs onto V1 ChCs are monosynaptic (*Cruikshank et al., 2010*). Repeated optogenetic stimulation (20 Hz) resulted in synaptic depression (*Figure 1I*), indicating RSC synapses may have a high release probability (*Zucker and Regehr, 2002*).

Finally, we performed paired recordings of L2/3 PyCs and ChCs to test their local connectivity within V1 (*Figure 1J–K*). Inducing APs in ChCs generated postsynaptic responses in ~45% (5/11) of PyCs. However, none of the 11 ChCs we recorded from responded to local PyC stimulation (*Figure 1J–K*), indicating a highly non-reciprocal connectivity motif. It also indicated that only few if any local L2/3 PyCs provide synaptic input to ChCs. It should be noted that the use of whole-cell

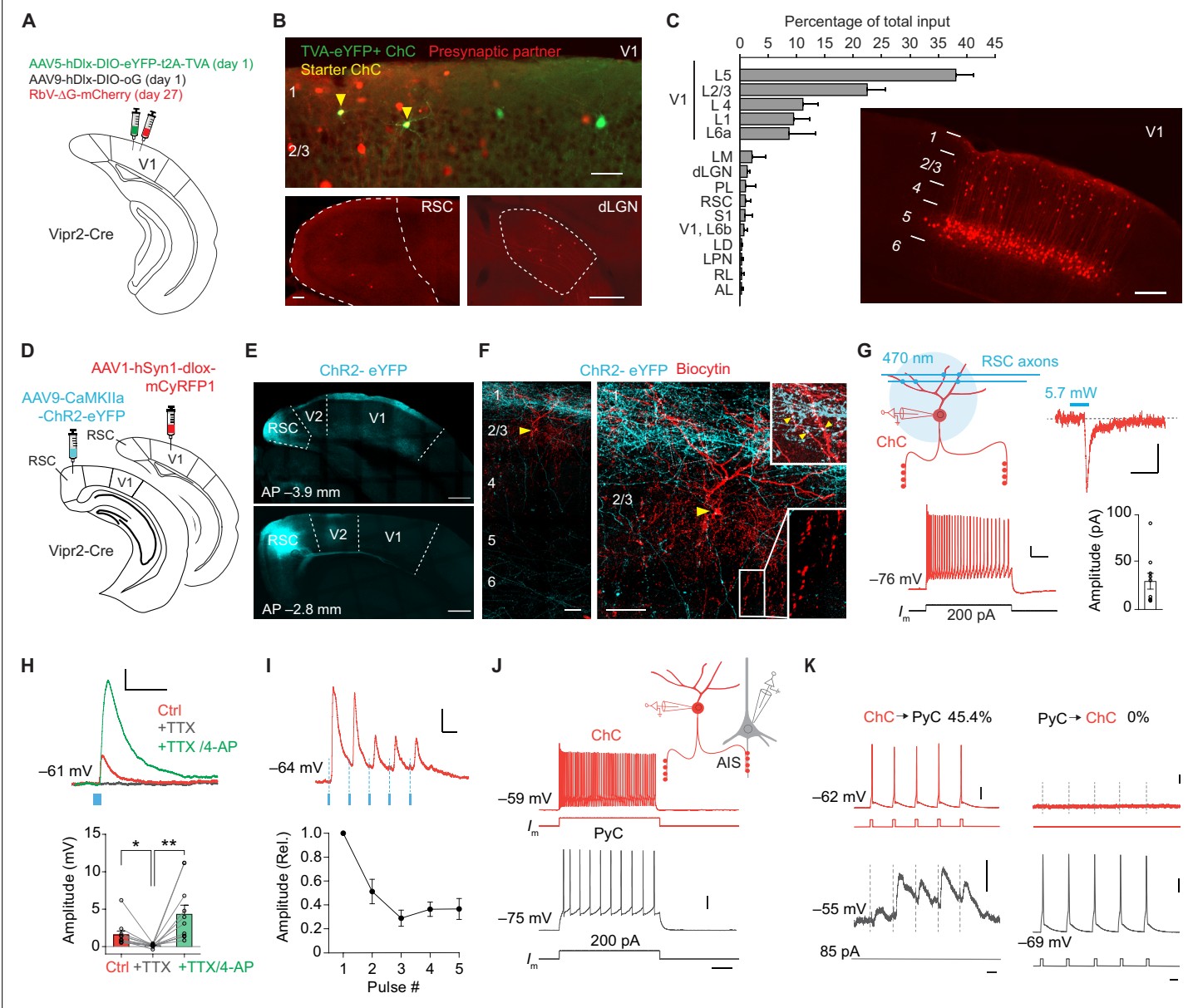

**Figure 1.** Chandelier cells (ChCs) receive input from L5 pyramidal cells (PyCs) and innervate L2/3 PyCs. (**A**) Schematic with viral strategy for selective monosynaptic retrograde rabies tracing of L2/3 ChCs. (**B**) Overview of superficial V1 region (top) with starter ChCs (yellow), non-starter ChCs (green), and presynaptic partners (red). Scale bar, 50 μm. Bottom: example images of RSC (left; scale bar, 100 μm) and dLGN (right; scale bar, 200 μm) containing input cells in red. Number of starter ChCs = 7.5 ± 3.8 (mean ± SEM, with a total of 30 starter ChCs from 4 mice). (**C**) Quantification of input sources to ChCs ($n$=4 mice) represented as percentage (mean ± SEM) of the total number of presynaptic neurons observed brain wide. LM, lateromedial visual area; dLGN, dorsal lateral geniculate nucleus; PL, posterolateral visual area; RSC, retrosplenial cortex; S1, primary somatosensory area; LD, lateral dorsal nucleus of the thalamus; LPN, lateral posterior nucleus of the thalamus; RL, rostrolateral area; AL, anterolateral visual area. The image shows the distribution of input neurons selectively within V1. Scale bar, 200 μm. (**D**) Schematic with viral strategy for optogenetic activation of RSC inputs to L2 ChCs. PyCs in RSC were labeled with ChR2-eYFP, ChCs in V1 were labeled with the red fluorophore mCyRFP1. (**E**) Confocal images showing the ChR2-eYFP (cyan) injection location in RSC (bottom) and their projections to L1 in V1 (top). Scale bar, 500 μm. (**F**) Confocal images of the biocytin fill (red) of mCyRFP+neurons revealed L2 ChC identity. Insets depict putative RSC inputs on apical dendrites of ChC in layer 1 (cyan, top) as well as characteristic rows of ChC bouton cartridges (bottom). Yellow arrow indicates soma. Scale bars, 50 μm. (**G**) Schematic of whole-cell patch-clamp recordings from mCyRFP+neurons. Current injections evoked firing patterns characteristic of ChCs. Scale bars, 10 mV, 100 ms. Optogenetic activation of RSC boutons evoked inward currents of on average 29.8 pA ($n$=11/13 ChCs from 13 slices in 5 mice). Bar shows mean and SEM, dots represent individual cells. Scale bars, 20 ms, 10 pA. (**H**) Tetrodotoxin (TTX)/4-aminopyridine (4-AP) bath application confirmed monosynaptic RSC (470 nm optogenetically evoked, blue) inputs in ChCs. RM ANOVA **p=0.0035, Holm-Šídák's multiple comparisons test, *p=0.012, **p=0.008. Bar shows mean ± SEM, dots represent individual cells, $n$=11 cells from 11 slices in 5 mice. Scale bars, 1 mV, 100 ms. (**I**) Optogenetic stimulation at 20 Hz revealed a reduction in postsynaptic

*Figure 1 continued on next page*

*Figure 1 continued*

potential amplitudes. Circles show mean ± SEM. Scale bars, 2 mV, 50 ms. *N*=8 cells from 8 slices in 3 mice. (**J**) Voltage responses to a current injection steps in ChCs and PyCs during simultaneous recordings. Scale bars, 100 ms, 20 mV. (**K**) Action potentials were generated by brief current injections in ChCs (left) or PyCs (right). In *n*=5 out of 11 pairs, ChC stimulation generated postsynaptic responses in PyCs. In *n*=0/11 PyC were projecting back onto ChC. Scale bars 10 ms, 20 mV, and 0.5 mV for subthreshold responses, 11 pairs from 11 slices in 6 mice.

The online version of this article includes the following figure supplement(s) for figure 1:

**Figure supplement 1.** Morphology of labeled chandelier cells (ChCs) and putative GABAergic input neurons in L2/3.

recordings with high chloride internal solution as performed here precludes determining whether ChC input on PyCs causes hyperpolarization or depolarization. These findings suggest that the rabies-labeled L1–4 neurons providing monosynaptic input to ChCs may include many inhibitory neurons, in line with previous work showing that V1 ChCs receive local input from L2/3 SST+ interneurons as well as neurogliaform cells (L2/3 NGCs), but not from L2/3 PyCs (*Jiang et al., 2015*). This is further supported by the distributed localization of the neurons labeled by the rabies virus (mCherry⁺), their presence in L1 and the observation that they lacked spines and that the soma appeared non-pyramidal (*Figure 1—figure supplement 1D–F*). Taken together, these results show that ChCs in V1 receive substantial input from local L5 PyCs, inhibitory neurons and long-range sources, while they locally innervate L2/3 PyCs.

## ChCs are modulated by arousal and show high correlations

Having studied their connectivity, we next looked at the in vivo response properties of ChCs and L2/3 PyCs using two-photon calcium imaging in awake animals. We injected adult Vipr2-Cre mice with an AAV vector driving expression of GCaMP6f under the control of a short CaMKIIa promoter to label putative PyCs and a Cre-dependent AAV vector driving expression of mRuby2-GCaMP6f to label ChCs in V1. We implanted the mice with a cranial window and head ring to allow head fixation on a running wheel (*Figure 2A*).

We first assessed activity of ChCs and L2/3 PyCs during spontaneous behavior by tracking running speed and pupil size while mice were viewing a uniform gray screen (*Figure 2A*, *Figure 2—video 1*). We restricted our field of view to the border between L1 and L2/3, where superficial ChCs are positioned. In line with earlier work, correlation analysis between calcium activity and both running speed and pupil size revealed that ChCs were mostly active during states of high arousal, more so than L2/3 PyCs (*Figure 2B–C*, multilevel statistical analyses were performed using a linear mixed effects model [LMEM] to account for dependencies in the data and variance between mice, see Materials and methods) (*Schneider-Mizell et al., 2021*; *Bugeon et al., 2022*). In addition, ChCs within the same field of view were highly correlated with each other, much more so than L2/3 PyCs (*Figure 2B–C*), suggesting that ChCs distribute a synchronized signal during high arousal.

To test whether ChCs also responded to visual stimuli, we examined their orientation and direction tuning by showing mice 1 s moving oriented gratings (*Figure 2D*, *Figure 2—video 2*). Although both ChCs and L2/3 PyCs had strong visual responses, ChCs were weakly tuned and showed a lower orientation selectivity index (OSI) than L2/3 PyCs (*Figure 2E–G*). The direction selectivity index (DSI) was similar between cell types.

## ChCs are weakly selective to visual information

The highly synchronized ChC activity, its correlation with arousal, and the relatively weak orientation tuning of ChCs suggested that while they signal behaviorally relevant events, they may only weakly encode visual stimulus features. To assess this more thoroughly, we assessed the visual response properties of ChCs. Artificial visual stimuli such as oriented gratings can reveal tuning to isolated stimulus parameters. However, experimental constraints limit the number of receptive field properties that can be tested this way. In addition, due to the nonlinear response selectivity of visual neurons, receptive field properties defined using gratings do not always generalize well to natural vision (*Felsen and Dan, 2005*). Therefore, to obtain a more complete picture of the cells' visual receptive field properties, we used natural stimuli in combination with a pre-trained deep convolutional neural network (CNN) to model single-cell visual responses and visualize their putative most exciting inputs (MEIs): the visual stimuli that elicit the strongest response in each individual neuron (*Cadena et al., 2019*; *Walker et al., 2019*; *Bashivan et al., 2019*; *Papale et al., 2021*; *Figure 3*). For this experiment, we

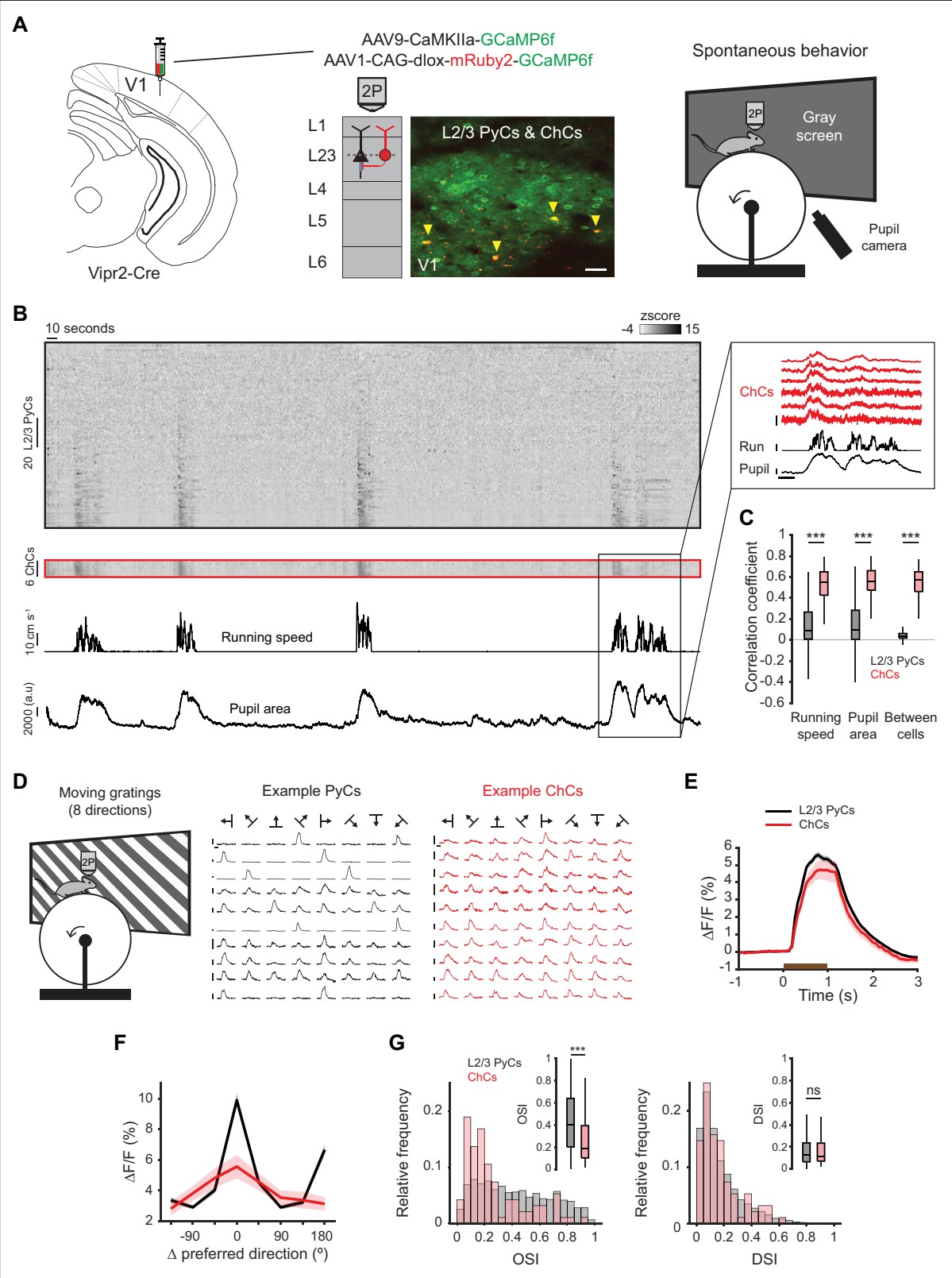

**Figure 2.** Chandelier cells (ChCs) are modulated by arousal and show high correlations. (**A**) Schematic of approach. ChCs (selectively expressing Cre in Vipr2-Cre mice) were identified using the red fluorophore mRuby2 and neuronal activity of ChCs (yellow arrows) as well as L2/3 pyramidal cells (PyCs) was tracked using two-photon calcium imaging of GCaMP6f. Scale bar, 50 μm. Mice were allowed to freely run or rest while viewing a gray screen. (**B**) Example recording of PyCs (black box) and ChCs (red box) with tracking of running speed and pupil area (bottom two rows). Single-cell ΔF/F traces

*Figure 2 continued*

were z-scored for display purposes. The inset highlights the correlated activity of six ChCs. Inset vertical scale bars, 4 z-score (ChCs), 10 cm/s (run), 2000 a.u. (pupil). Horizontal scale bar, 10 s. (**C**) Average correlation coefficients for Δ*F/F* of PyCs and ChCs with running speed, pupil area, and within cell type. ChCs show higher correlation coefficients than PyCs across conditions (15 sessions from 8 mice, *n*=1883 PyCs and 95 ChCs, 19 ± 2.93 ChC pairs per field of view). Linear mixed effects model (LMEM) for all comparisons. ***: p<0.001, ns: not significant. Box plots represent median, quantiles, and 95% confidence interval (CI) over neurons. (**D**) Schematic of recording during visual stimulation with moving gratings (left) and responses to all directions, e.g., PyCs (middle) and ChCs (right). Vertical scale bars, 10% Δ*F/F*, horizontal scale bars, 1 s. (**E**) Average response of L2/3 PyCs and ChCs to moving gratings (1 s, brown bar). (**F**) Orientation and direction tuning curves. Curves represent mean ± SEM over neurons after aligning single-cell curves to their preferred direction. (**G**) Histograms and average orientation selectivity index (OSI) and direction selectivity index (DSI) (insets) for L2/3 PyCs and ChCs. ChCs are more weakly tuned to orientation, but not direction of moving gratings than L2/3 PyCs.

The online version of this article includes the following video(s) for figure 2:

**Figure 2—video 1.** Example two-photon calcium imaging recording during spontaneous behavior.

https://elifesciences.org/articles/91153/figures#fig2video1

**Figure 2—video 2.** Example two-photon calcium imaging recording with visual stimulation.

https://elifesciences.org/articles/91153/figures#fig2video2

used two new groups of mice. In one group, we injected Vipr2-Cre mice with an AAV vector driving expression of GCaMP8m (*Zhang et al., 2023*) (to label L2/3 PyCs and ChCs) and a Cre-dependent AAV vector driving expression of mCyRFP1 (to label ChCs). Since our rabies tracing results revealed that V1 L5 PyCs were the most abundant source of synaptic input to ChCs, we targeted L5 PyCs in a second group of mice. We selectively labeled L5 PyCs using a tail vein injection of a Cre-dependent PhP.eB-serotyped (*Deverman et al., 2016*) AAV vector driving expression of GCaMP6f in Rbp4-Cre mice (*Gerfen et al., 2013*; *Figure 3A*). We then recorded neural activity in both groups while mice were shown a set of 4000 natural images, with 40 of these images being shown 10 times each. We used the neural responses of individual neurons to determine their selectivity, and subsequently, to optimize a CNN and obtain an estimate of the cells' MEIs (*Figure 3A*).

We first focused on properties derived from recorded neuronal responses (*Figure 3B–E* and *Figure 3—figure supplement 1A*). ChCs responded strongly to natural images, comparable to L2/3 PyCs and L5 PyCs (*Figure 3B*). In order to test the selectivity of neurons for specific natural stimuli, we made use of the subset of 40 images that we presented 10 times. For each cell, we sorted the average responses to these images based on their strength, creating a ranked distribution that revealed differences in stimulus selectivity between cell types (*Figure 3C*). L2/3 PyCs and L5 PyCs responded strongly to only a few images, indicating high selectivity. In contrast, ChCs were weakly selective as shown by their strong responses to many images. We quantified image selectivity for each neuron by calculating sparsity (*Zoccolan et al., 2007*). High sparsity indicates strong responses to only a few images, while low sparsity indicates equal responsiveness to many images. As evident from their flattened curve in the ranked distribution, ChCs had significantly lower sparsity than L2/3 PyCs and L5 PyCs (*Figure 3C*, inset). Next, we reasoned that if ChCs are weakly selective for visual stimuli, their between-cell correlation should remain high even during exposure to variable visual input. Indeed, correlations between ChCs were considerably higher than those for L2/3 PyCs and L5 PyCs (*Figure 3D*), similarly to the situation during spontaneous behavior (*Figure 2C*). Finally, to test whether ChC activity contained less information about the visual stimuli than L2/3 PyC activity, we performed population decoding on the 40 images using linear discriminant analysis (LDA). We compared decoding accuracy using all ChCs (*n*=34) with that of a distribution of accuracies obtained from randomly subsampling equal numbers of L2/3 PyCs (see Materials and methods). Decoding accuracy of ChCs was significantly lower than that of L2/3 PyCs (*Figure 3E*). Together, these results show that ChCs are visually responsive, but weakly selective to visual information.

While orientation tuning and sparsity are useful measures of selectivity, they do not provide information about the type of stimuli that excite the neurons most strongly. To determine the MEI of each neuron, we used the responses of individual ChCs, L2/3 PyCs, and L5 PyCs to optimize a pre-trained CNN (*Cadena et al., 2019*; *Walker et al., 2019*; *Bashivan et al., 2019*; *Papale et al., 2021*). First, we obtained predicted (artificial) responses from the pre-trained CNN to a batch of the natural images. We then fit a mapping function from predicted responses to neuronal responses (recorded from the mice). The mapping function consisted of a set of spatial weights to model the location and spatial extent of the RF and a set of feature weights to model the feature selectivity (e.g. orientation) of each

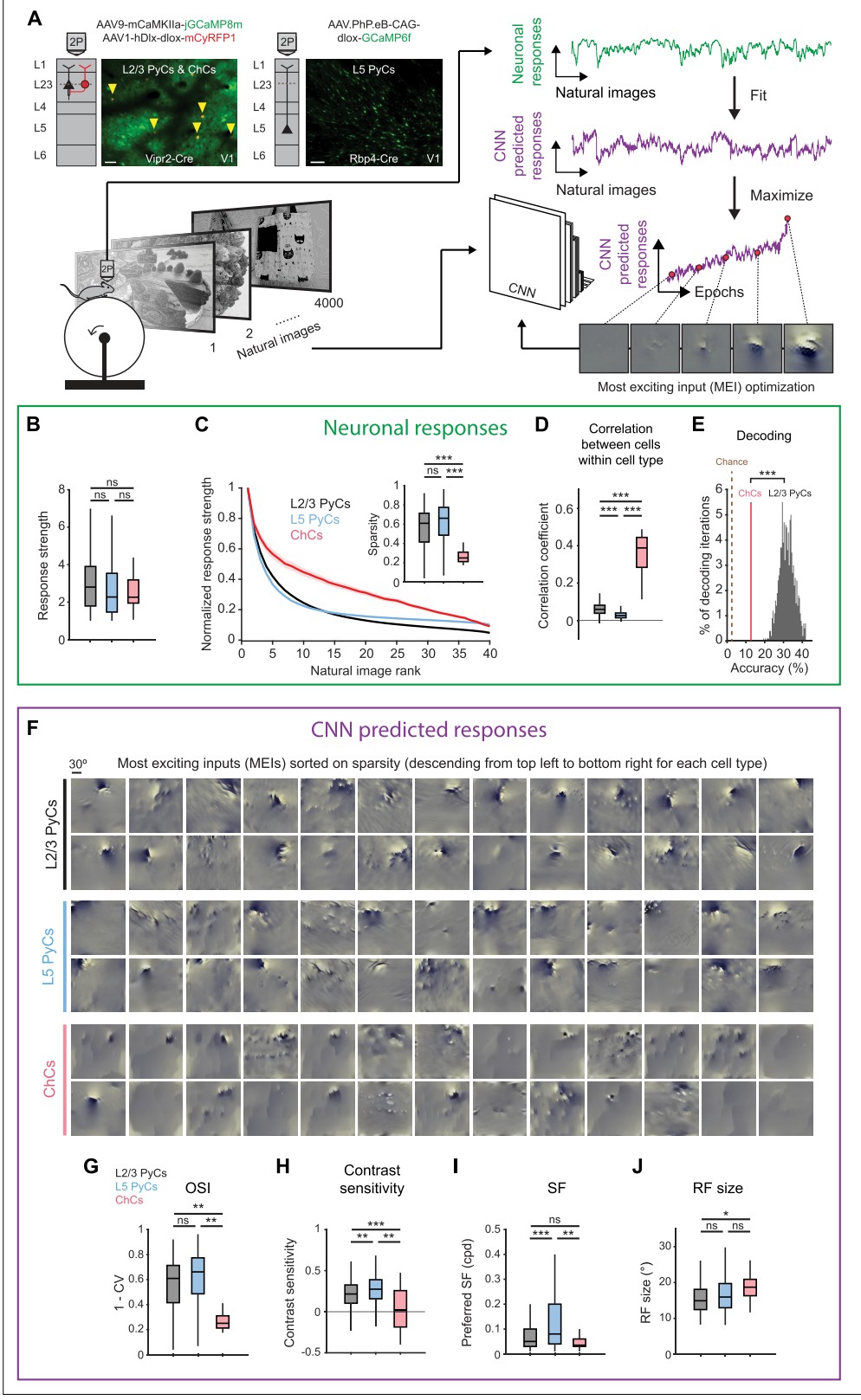

**Figure 3.** Chandelier cells (ChCs) are weakly selective to visual information. (**A**) Schematic of experiment and convolutional neural network (CNN) model fitting. Mice expressing GCaMP8m in L2/3 pyramidal cells (PyCs) and ChCs (Vipr2-Cre mice) or GCaMP6f in L5 PyCs (Rbp4-Cre mice) were shown a set of 4000 images. We trained a CNN to predict single-cell responses to a range of visual stimuli and to derive most exciting inputs (MEIs). Traces

*Figure 3 continued on next page*

*Figure 3 continued*

(right) represent average responses to natural images (green) and the activity predicted by the CNN (purple) for an example neuron. Scale bars example field of view, 50 µm. (**B**) Average response strength to natural images for different neuronal cell types. Box plots represent median, quantiles and 95% confidence interval (CI) over neurons. n=1015 L2/3 PyCs, 1601 L5 PyCs and 34 ChCs. LMEM for all comparisons, ***: p<0.001, **: p<0.01, *: p<0.05, ns: not significant. (**C**) Average normalized response strength for different neuronal cell types on a subset of 40 natural images (compared to baseline, see Materials and methods). Images are ranked on the strength of the response they elicited for each neuron. ChCs curves are flatter than L2/3 PyCs and L5 PyCs, indicating lower stimulus selectivity. Inset: as in B, but for sparsity (a measure for stimulus selectivity). ChCs have lower sparsity than L2/3 PyCs and L5 PyCs. (**D**) As in B, but for correlation during visual stimulation. ChCs have higher within cell type correlations than L2/3 PyCs and L5 PyCs (21 ± 5.08 ChC pairs per field of view, mean ± SEM). (**E**) Natural image decoding accuracy for ChCs and L2/3 PyCs. ChC decoding accuracy (red line, 12.55%) was significantly lower than a distribution of decoding accuracies performed using equal numbers of subsampled L2/3 PyCs. Permutation test, ***p<0.001. The brown dotted line indicates theoretical chance level (2.5%). (**F**) Single-cell MEIs sorted by response sparsity (highest 26 neurons, descending from top left to bottom right). Note the diffuse and unstructured patterns in ChC MEIs. (**G**) As in B, but for orientation selectivity (OSI). ChCs have lower OSI than L2/3 PyCs and L5 PyCs. (**H**) As in B, but for contrast sensitivity. ChCs have lower contrast sensitivity than L2/3 PyCs and L5 PyCs. (**I**) As in B, but for spatial frequency (SF) tuning. ChCs prefer lower SFs than L5 PyCs. (**J**) As in B, but for receptive field (RF) size. ChCs have bigger RFs than L2/3 PyCs.

The online version of this article includes the following figure supplement(s) for figure 3:

**Figure supplement 1.** Convolutional neural network (CNN) model performance and tuning curves obtained from CNN predictions.

neuron. To fit the model, we compared the neuronal responses with the predicted responses made by the CNN. We repeated this process several times using different batches of natural images. On each iteration, we changed both the spatial and feature weights of the model to minimize the error between neuronal responses and predicted responses, until no further improvements were made. The result of this optimization was the final model, comprising an artificial copy of each individual neuron that could be used to predict visual responses which are highly representative of the neuron's visual response properties (see Materials and methods and *Figure 3—figure supplement 1B–C*).

The CNN allowed us to obtain an MEI for each neuron by presenting artificial visual stimuli (*Figure 3F*, see Materials and methods) (*Walker et al., 2019*). MEIs can reveal complex nonlinear RF properties such as corners, curves, and textures that represent optimal visual input, which is otherwise difficult to quantify in a single metric. Inspection of the MEIs and quantification of predicted responses to simple artificial stimuli revealed striking differences between cell types. For instance, MEIs of L2/3 PyCs often displayed clearly oriented edge-like patterns with sharp ON and OFF regions, which were much less apparent in ChCs (*Figure 3F*), while L5 PyCs showed a mixed form of selectivity, including both edge-like patterns as well as more complex textures. In line with this observation and in agreement with our orientation tuning experiments (*Figure 2D–E*), the modeled ChCs had lower OSIs than L2/3 PyCs and L5 PyCs (*Figure 3G* and *Figure 3—figure supplement 1D*). Furthermore, ChC MEIs mostly lacked high-contrast patterns, containing high spatial frequencies (SFs). Accordingly, the quantification of contrast tuning and SF of modeled neurons revealed that ChCs were less contrast tuned than L2/3 PyCs and L5 PyCs (*Figure 3H* and *Figure 3—figure supplement 1E*) and preferred lower SFs than L5 PyCs (*Figure 3I*), which might be related to the interdependence between contrast sensitivity and SFs (*Heimel et al., 2010*; *Boynton et al., 1999*). The smooth and featureless MEIs of ChCs were further reflected by their larger RFs (*Figure 3J*).

Finally, given that ChCs receive most of their inputs from local L5 PyCs (*Figure 1C*), we asked whether ChC MEIs could be the result of combinations of L5 PyC inputs. We generated MEIs to maximize the response of combinations of L5 PyCs and found that many of the resulting MEIs were less structured and lacked clearly oriented edge-like patterns (*Figure 3—figure supplement 1F*). The similarity of these MEIs with those we found for ChCs is in line with the idea that ChCs are driven by input from large number of L5 PyCs (but does not exclude alternative explanations). Together with the observation that the activity of ChCs is strongly influenced by non-visual factors such as arousal, these results support a view in which ChCs are visually responsive but mostly invariant to the spatial and featural arrangement of visual stimuli.

## Activity of V1 ChCs and L2/3 PyCs in mice in a virtual tunnel paradigm

To determine how the interaction between behavior and visual input drives the activity of ChCs, we recorded their calcium responses in V1 of mice in a virtual tunnel paradigm. We designed the tunnel in a way that allowed us to examine L2/3 PyC and ChC activity in response to multiple variables, such as visual stimuli, locomotion, and visuomotor mismatch (errors between expected and perceived visual input) (*Figure 4A, B*). In the first, 1-m-long section of the tunnel ('visual section'), two visual patterns (grating and checker) were repeated three times on a white noise background (*Figure 4B*, left). The second section ('non-visual') immediately followed the visual section. It consisted of an even, gray area and included an auditory cue predicting a reward (cue at $t=1$ s, reward at $t=3$ s) and a 6 s waiting period before the next trial started (*Figure 4B*, right). Mice were trained in a minimum of six training sessions, during which they became acquainted to the tunnel and learned to lick for a reward. After training, mice performed the task twice under the two-photon microscope, allowing us to record 1256 PyCs and 38 ChCs from a total of 12 locations in 6 mice (*Figure 4C*).

ChCs were highly active at the onset of the visual section of the tunnel but became suppressed when the visual stimuli became visible to the mice (*Figure 4C–D* and *Figure 4—figure supplement 1A–B*). Interestingly, while a large subpopulation of PyCs were activated by the visual stimuli, many others showed a similar activity profile as ChCs, suggesting functionally separate populations of PyCs (*Figure 4C*). In order to test this, we performed hierarchical clustering on z-scored average activity in the visual section of the tunnel. We used silhouette analysis to examine the separation distance between clusters for different numbers of clusters, which showed that the optimal number of clusters was two (*Figure 4—figure supplement 1C*). Examination of the two clusters of PyCs revealed that they had opposite activity patterns. Cluster 1 was activated by visual stimuli but suppressed in the non-visual section, while cluster 2 was suppressed by visual stimuli but activated in the non-visual section, like ChCs (*Figure 4C–D*). Thus, we named the PyCs in these clusters 'visually responsive PyCs' (V-PyCs) and 'non-visually responsive PyCs' (NV-PyCs) respectively.

Although the difference between the two clusters was striking, silhouette evaluation cannot rule out a lack of functional clustering: i.e., where the true number of clusters is one. Therefore, to test whether our data were better described by two clusters or one, we compared the separability of the visual response distributions obtained for the real V-PyCs and NV-PyCs to the separability of 1000 random combinations using a permutation test (*Figure 4—figure supplement 1C–F*, see Materials and methods). The separability between V-PyCs and NV-PyCs was higher than expected by chance (*Figure 4—figure supplement 1F*), indicating that V-PyCs and NV-PyCs are functionally separate clusters in our tunnel paradigm.

## Visuomotor mismatch responses in ChCs and non-visual PyCs

Previous research has demonstrated that a significant fraction of V1 PyCs exhibit strong visuomotor mismatch responses when the visual flow of the tunnel is abruptly stopped while the mice are still running (*Attinger et al., 2017*; *Jordan and Keller, 2020*; *Keller et al., 2012*; *Zmarz and Keller, 2016*; *Jordan and Keller, 2023*; *Muzzu and Saleem, 2021*; *O'Toole et al., 2023*). Recent evidence suggests that PyCs with visuomotor mismatch responses may belong to a genetically distinct subpopulation that is less visually responsive (*O'Toole et al., 2023*). Therefore, we assessed whether the two populations of PyCs we identified also differed in terms of their visuomotor mismatch responses. In a subset of trials, we briefly interrupted the visual flow for 0.5 s while the mice were running, to create a visuomotor mismatch. NV-PyCs displayed stronger responses to both the visuomotor mismatch and the onset of running compared to the V-PyCs (*Figure 4E–F, I–J*). These findings support the idea that different populations of PyCs in L2/3 exist, one primarily responding to visual stimuli and the other to locomotion and visuomotor mismatch.

ChCs showed a similar but more pronounced activity pattern compared to NV-PyCs: they were active at the start of the tunnel, while the visual stimuli in the tunnel suppressed their activity (*Figure 4C, D, G, and H*). Like NV-PyCs, ChCs were mostly driven by locomotion (*Figure 4F*) and showed strong responses to visuomotor mismatch (*Figure 4E, I*). In line with this, we found that ChCs were more strongly correlated with NV-PyCs than with V-PyCs (*Figure 4K*).

It is known that a subpopulation of ChCs express parvalbumin (PV) (*Taniguchi et al., 2013*). Indeed, ChCs in adult V1 can be selectively labeled using the combination of markers Vipr2 and PV (*Tasic et al., 2018*; *Bugeon et al., 2022*), but it is not known whether PV+-ChCs and Vipr2-ChCs (which

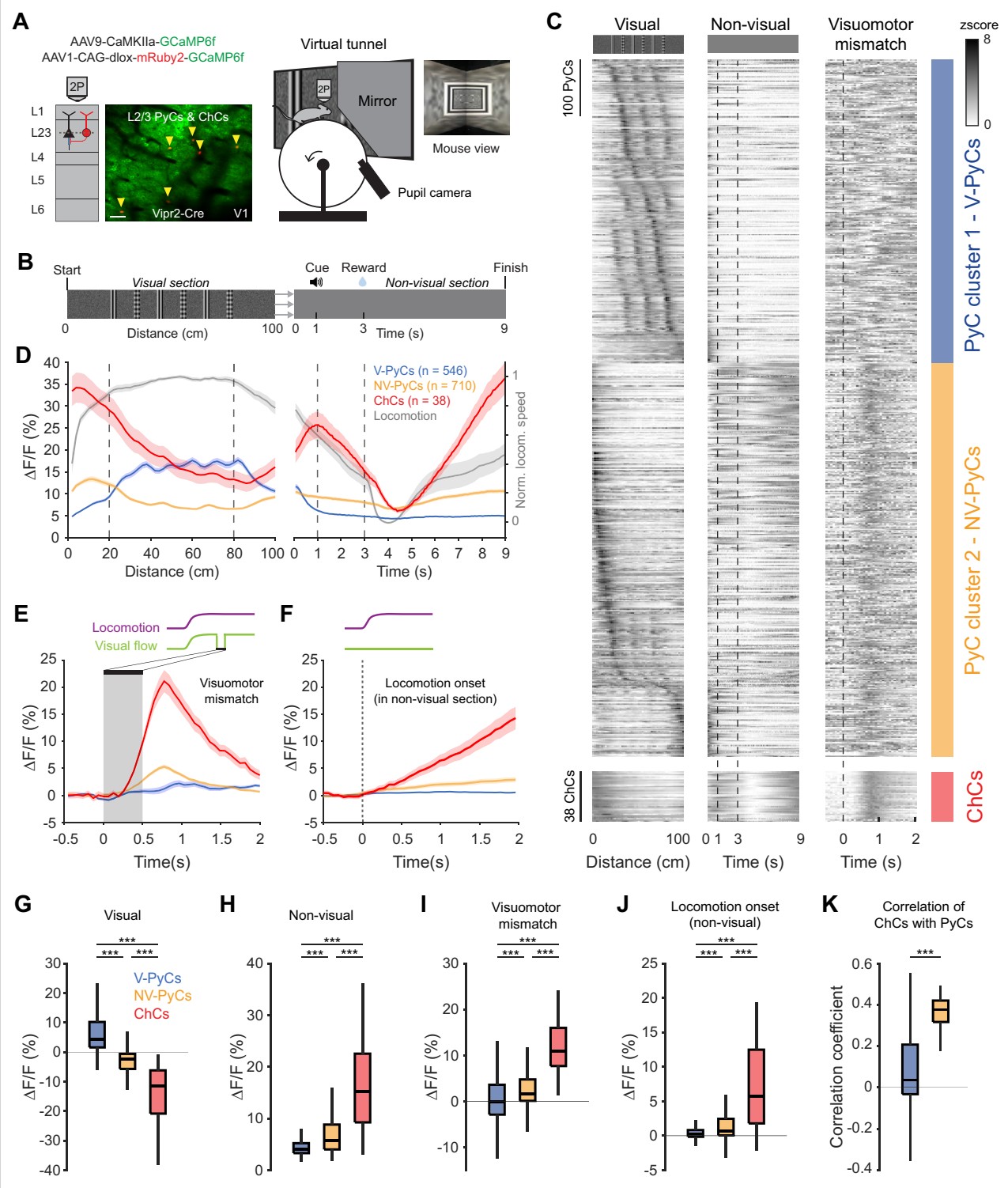

**Figure 4.** Locomotion and visuomotor mismatch drive chandelier cell (ChC) activity in a virtual tunnel. (**A**) Schematic of approach. Vip2r-Cre mice were head-fixed on a running wheel in a visual virtual tunnel. ChCs were identified using the red fluorophore mRuby2 and neuronal activity of ChCs (yellow arrows) as well as pyramidal cells (PyCs) was tracked using two-photon calcium imaging of GCaMP6f. Scale bar, 50 μm. (**B**) Virtual tunnel design. Mice ran through a virtual tunnel consisting of a 1-m-long visual section (containing visual stimuli along the walls) immediately followed by a non-visual reward zone in gray screen conditions. In the non-visual section, an auditory cure predicted a water reward 2 s later. After a 6 s timeout the next trial started. (**C**) Single-cell z-scored average activity of all PyCs (blue/orange) and ChCs (red) during visual, non-visual, and visuomotor mismatch parts of the tunnel (*n*=1256 PyCs and 38 ChCs, 12 sessions from 6 mice). PyCs were clustered in two populations using hierarchical clustering based on their z-scored

*Figure 4 continued on next page*

*Figure 4 continued*

activity in the visual section. Cells are sorted on cluster followed by peak activation location. Note the difference in activity between the visual (cluster 1: V-PyCs, blue) and non-visual (cluster 2: NV-PyCs, orange) PyC populations. (**D**) Average population traces of V-PyCs, NV-PyCs, and ChCs in the visual (left) and non-visual (right) part of the tunnel. Normalized locomotion speed (right y-axis) is depicted in gray. The ChCs follow the activity profile of NV-PyCs. Traces represent mean ± SEM over neurons for $\Delta F/F$ and mean ± SEM over sessions for locomotion speed. (**E**) Average population traces during visuomotor mismatch events. NV-PyCs and ChCs show strong mismatch responses. (**F**) Average population traces at locomotion onset in the non-visual section of the tunnel. NV-PyCs and ChCs show strong locomotion onset responses. (**G**) Average activity during visual stimuli (20–80 cm) compared to start of tunnel (0–20 cm). NV-PyCs and ChCs were strongly suppressed by visual stimuli. Box plots represent median, quantiles, and 95% confidence interval (CI) over neurons. LMEM for all comparisons, ***: p<0.001. (**H**) As in G, but for activity in the entire non-visual part of the tunnel. (**I**) As in G, but for visuomotor mismatch events. (**J**) As in G, but for locomotion onset events. (**K**) Average correlation coefficient of ChCs with PyCs in visual and non-visual section of the tunnel. ChCs are more strongly correlated with NV-PyCs than V-PyCs.

The online version of this article includes the following figure supplement(s) for figure 4:

**Figure supplement 1.** Open loop and PV⁺ chandelier cell (ChC) responses.

includes both PV⁺ ChCs and PV⁻ ChCs) are functionally different. We therefore tested whether PV⁺ ChCs had similar response properties as Vipr2-ChCs by repeating our tunnel experiments using two Vipr2-Cre X PV-FlpO × AI65(RCFL-tdTom) (*Daigle et al., 2018*) mice (*Figure 4—figure supplement 1G*). In these mice, tdTomato was only expressed in cells expressing both Vipr2 and PV. We found that PV⁺-ChCs showed identical activity patterns as Vipr2-ChCs (*Figure 4—figure supplement 1H–P*). In addition, in this smaller sample of mice we found the same separation of V-PyCs and NV-PyCs as described in *Figure 4*.

Finally, we assessed the activity of ChCs when visual flow was uncoupled from the running speed of the mouse (open loop condition). This revealed that responses to sudden halts in visual flow that were independent of locomotion were much weaker than responses to closed loop visuomotor mismatch (*Figure 4—figure supplement 1Q*). In addition, open loop onset of visual flow when the mouse was not running resulted in a suppression of ChC activity (*Figure 4—figure supplement 1Q*). We conclude that in mice trained in our virtual tunnel paradigm, ChCs responded predominantly to locomotion and visuomotor mismatch when visual flow stopped during running, while they were suppressed by visual stimuli or when visual flow started while mice were stationary.

## Experience-dependent visual plasticity of ChCs and NV-PyCs

The observation that ChCs are suppressed by visual stimulation in the virtual tunnel was unexpected, as our grating (*Figure 2*) and natural image (*Figure 3*) experiments showed that ChCs are activated by oriented moving gratings and natural images. However, our passive viewing experiments were performed in mice largely naive to visual stimulation. In contrast, mice behaving in our tunnel paradigm were repeatedly exposed to visual stimulation during the tunnel training phase before we started recording neural activity during behavior. We therefore hypothesized that the observed suppression of ChCs by visual stimulation in the tunnel was caused by experience-dependent plasticity induced by the repeated exposure to visual stimulation during training. To test this, we assessed visual responses in the same neurons to moving oriented gratings before and after training in the virtual tunnel, while mice were passively viewing the stimuli.

In naive mice, both PyCs and ChCs responded strongly to visual stimulation. Interestingly, the same ChCs showed weak or even suppressed visual responses after mice had been trained in the tunnel (*Figure 5A–D*). In contrast, the response strength of PyCs was only mildly and statistically non-significantly reduced after training (*Figure 5D*). The OSI and DSI were unchanged after training (PyC OSI: p=0.1316, PyC DSI: p=0.9573, ChC OSI: p=0.2199, ChC DSI: p=0.9774, *Supplementary file 1*). Since ChCs showed similar activity patterns in the tunnel as NV-PyCs, we asked whether this subpopulation also showed sensory-evoked plasticity. In a subset of mice, we matched neurons recorded in all three sessions (pre-training, tunnel, and post-training). We used the tunnel session to separate the PyCs in the same two functional clusters and then assessed their responses before and after training. This confirmed that visuomotor experience in the virtual tunnel significantly reduced responses in NV-PyCs (*Figure 5E–G*). Finally, as recent work showed that motor learning reduced correlated activity of ChCs in PFC (*Jung et al., 2023*), we tested whether visual experience had a similar effect in V1. Indeed, pairwise correlations of ChCs activity were also lower in mice trained in the virtual tunnel ($R$=0.54 ± 0.01 in naïve mice, $R$=0.46 ± 0.03 in trained mice, p<0.01). These results indicate that ChC

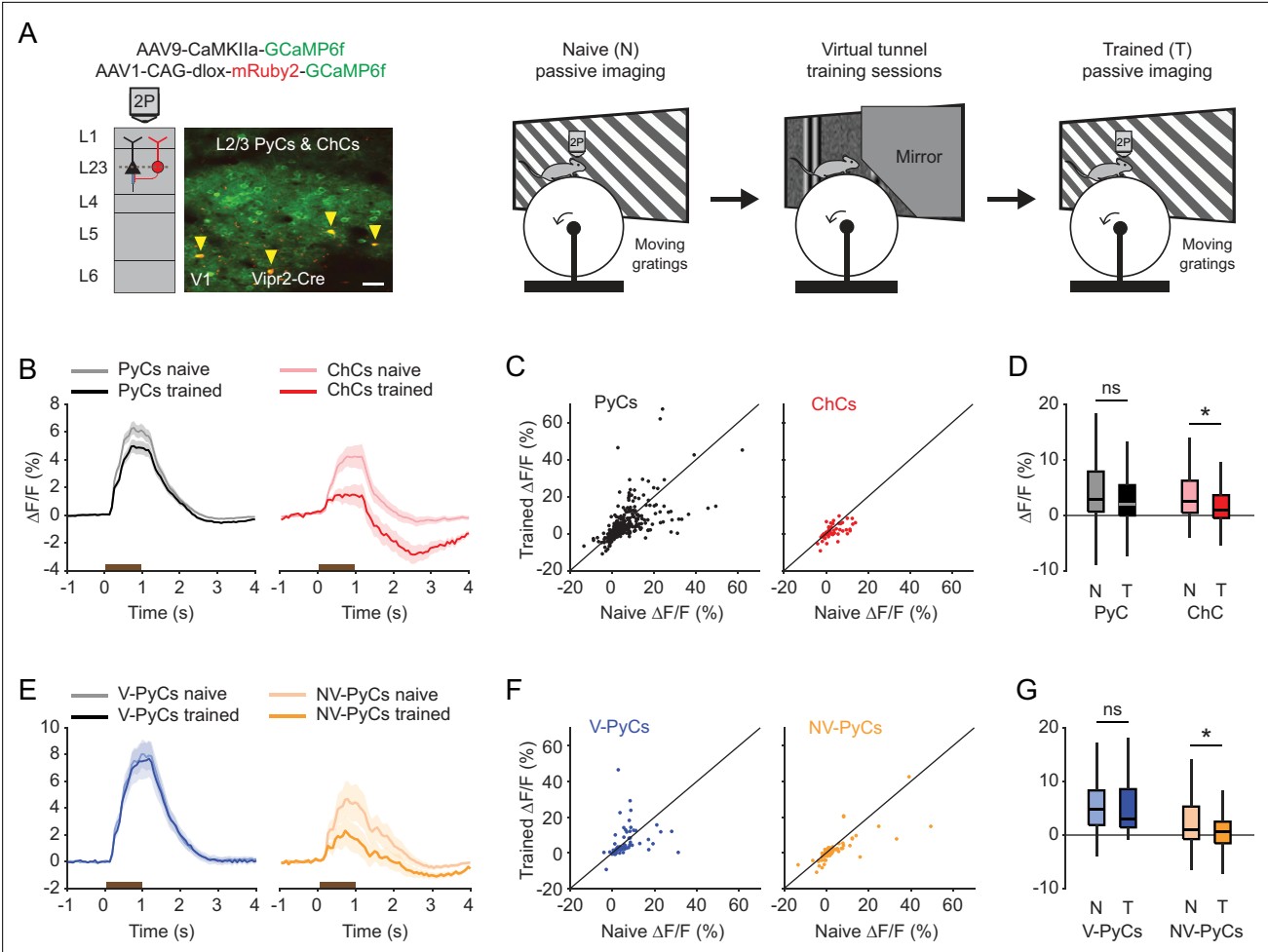

**Figure 5.** Experience-dependent visual plasticity of chandelier cells (ChCs) and non-visually responsive pyramidal cells (NV-PyCs). (**A**) Schematic of viral strategy and experimental setup. In the passive imaging sessions before and after training, mice were imaged while they passively viewed moving gratings. In-between those sessions mice were trained in the virtual tunnel where they were repeatedly exposed to visual stimuli. Yellow arrows point to ChCs. Scale bar, 50 μm. (**B**) Average population response traces to a 1 s (brown bar) moving grating for all PyCs and ChCs chronically imaged and matched in 'naive' and 'trained' sessions (*n*=411 PyCs and 51 ChCs, 8 sessions from 5 mice). Traces represent mean ± SEM over neurons. (**C**) Single-cell visual response magnitude pre and post training (averaged over 0.2–1.2 s after stimulus onset). (**D**) Average visual response magnitude in naive (N) and trained (T) conditions. ChCs, but not PyCs, show plasticity after training. LMEM for all comparisons, *: p<0.05, ns: not significant. Box plots represent median, quantiles, and 95% confidence interval (CI) over neurons. (**E**) As in B, but for all PyCs chronically imaged and matched in naive, tunnel, and trained sessions (*n*=68 V-PyCs and 71 NV-PyCs, 5 sessions from 3 mice). (**F**) As in C, but for PyC subtypes. (**G**) As in D, but for PyC subtypes. NV-PyCs show plasticity after training.

responses underwent sensory-evoked plasticity during the repeated visual exposure, even though the visual stimuli were different from those encountered during training in the virtual tunnel. This is in line with our observation that ChCs are weakly selective to visual stimuli.

## Visuomotor experience in the virtual tunnel is accompanied by plasticity of ChC-AIS connectivity

Previous work has shown that prolonged chemogenetic activation of ChCs or PyCs results in plasticity of ChC bouton numbers at the AIS (*Pan-Vazquez et al., 2020*). In addition, PyC activation also causes geometric plasticity of the AIS location and/or length (*Kuba et al., 2010*; *Jamann et al., 2021*; *Grubb and Burrone, 2010*; *Gutzmann et al., 2014*; *Wefelmeyer et al., 2015*). We therefore hypothesized that, in addition to the changes in response properties of ChCs, the visuomotor experience in the virtual tunnel paradigm may induce axonal plasticity of ChCs. To test this hypothesis, we used

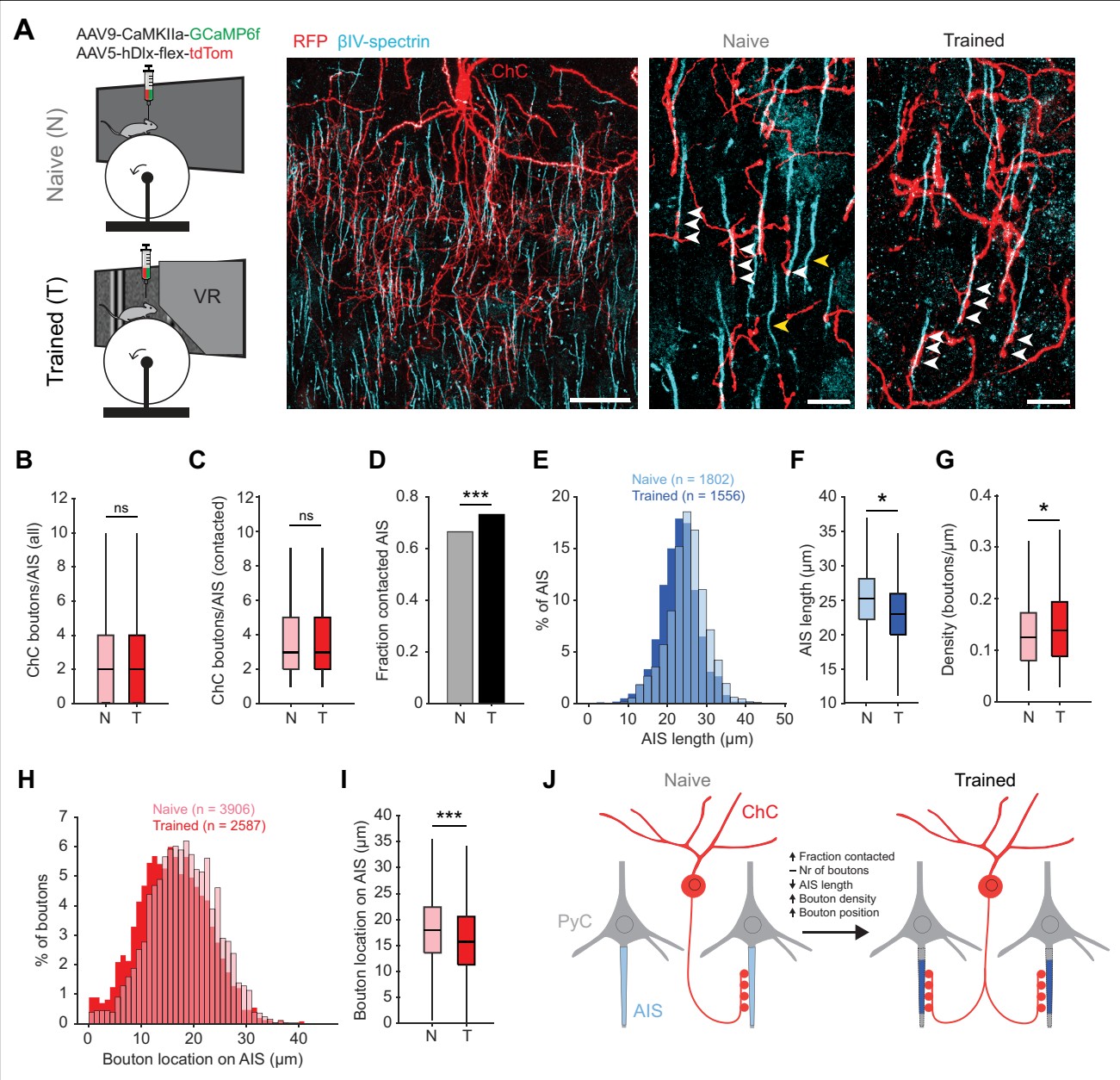

**Figure 6.** Training in the virtual tunnel induces plasticity of chandelier cell (ChC)-axon initial segment (AIS) connectivity. (**A**) Left: experimental design showing naive mice and mice trained in the virtual tunnel. Middle: confocal image showing V1 L2/3 after immunostaining to visualize ChC processes (α-RFP) and the AIS (α-βIV-spectrin). Scale bar, 30 μm. Right: example colocalization (white) of ChC (red) and AISs (cyan) in naive and trained mice. White arrows show putative ChC boutons on AISs. Yellow arrows point to uncontacted AISs. Scale bars, 10 μm. (**B**) The average number of boutons on all AISs is similar between naive and trained mice. Five naive and 3 trained mice, 2–3 slices per mouse, $n$=1802 AISs (N) and 1007 AISs (T). LMEM for all comparisons, ***: $p<0.001$, **: $p<0.01$, **: $p<0.05$, ns: not significant. Box plots represent median, quantiles, and 95% confidence interval (CI). (**C**) The number of boutons on AISs with at least 1 bouton is unaffected by training. Five naive and 3 trained mice, 2–3 slices per mouse, $n$=1200 AIS (**N**) and 738 AIS (**T**). (**D**) Fraction of cells contacted by at least 1 ChC bouton increases with training. $X^2(1)$=13.53, ***: $p<0.001$, 5 naive and 3 trained mice, 2–3 slices per mouse, $n$=1802 AIS (N) and 1007 AIS (T). (**E**) Histogram of AIS lengths in naive and trained mice. (**F**) AIS length is decreased by training. Five naive and 5 trained mice, 2–3 slices per mouse, $n$=1802 AISs (N) and 1556 AISs (T). (**G**) Bouton density on the AIS is increased by training. Five naive and 3 trained mice, 2–3 slices per mouse, $n$=1200 AIS (N) and 738 AIS (T). (**H**) Histogram showing absolute location of boutons on the AIS. (**I**) Average bouton location on the AIS. Boutons in trained mice are located more closely to the start of the AIS. Five naive and 3 trained mice, 2–3 slices per mouse, $n$=3906 boutons (N) and 2587 boutons (T). (**J**) Schematic model of the changes observed in trained mice vs naive mice.

immunohistochemistry on V1 slices to visualize L2/3 PyC AISs and boutons of tdTomato-labeled ChCs in naive mice and mice after virtual tunnel training (*Figure 6A*).

We first quantified putative ChC boutons on the AIS of L2/3 PyCs (*Figure 6A*). While the average number of virally labeled ChC boutons per AIS remained constant (~2–3 ChC boutons/AIS, *Figure 6B and C*), we observed that visuomotor experience increased the fraction of AISs being contacted by ChCs (*Figure 6D*). We also found that the AIS of PyCs in trained mice were shorter than in naive mice (*Figure 6E–F*), resulting in an increased density of ChC boutons (i.e. ChC boutons per µm AIS length, *Figure 6G*). Finally, we found that ChC boutons were localized more closely to the base of the AIS (*Figure 6H–I*).

Taken together, these results show that after the mice were trained in the visuomotor task, not only response properties of ChCs in V1 had changed, but the number of AISs with putative ChC contacts had increased and activity-dependent AIS shortening of their post-synaptic PyC targets took place (*Figure 6J*).

## ChCs weakly inhibit PyC activity independent of locomotion speed

There has been controversy on whether ChCs hyperpolarize or depolarize their postsynaptic PyC targets (*Pan-Vazquez et al., 2020*; *Szabadics et al., 2006*; *Woodruff et al., 2009*; *Woodruff et al., 2011*; *Woodruff et al., 2010*; *Glickfeld et al., 2009*). However, recent work in adult mice has reported inhibitory effects in prelimbic cortex, S1, and hippocampus (*Pan-Vazquez et al., 2020*; *Lu et al., 2017*; *Dudok et al., 2021*). In order to study postsynaptic effects of ChC activity on L2/3 PyCs in V1, we used inhibitory k-opioid receptor (KORD)-based chemogenetics (*Vardy et al., 2015*) to globally silence ChCs in V1. We expressed KORD-mCyRFP1 or tdTom (control group) in Vipr2-Cre mice along with GCaMP6f (*Figure 7A*). We then recorded 15 min of ChC and L2/3 PyC activity before and after injection of the KORD ligand Salvinorin B (SalB, 10 mg/kg, s.c.) while mice were allowed to freely run or rest in front of a gray screen. We first looked at the average activity of neurons in both sessions. Surprisingly, L2/3 PyCs showed only a mild increase in activity after SalB injection, even though ChC activity was strongly and significantly reduced (*Figure 7B*).

Considering the strong modulation of ChCs by arousal, we next looked at effects of ChC silencing on L2/3 PyC activity across different locomotion speeds. Binning activity based on locomotion speed again revealed the relationship between locomotion and ChC activity and to a lesser extent L2/3 PyC activity (*Figure 7C*). Although the locomotion modulation curve of ChCs decreased in both amplitude and steepness after SalB injection, only the amplitude was affected for PyCs. Indeed, a linear fit of activity as a function of locomotion speed before and after SalB injection confirmed that the intercept but not the slope of the fit was increased for PyCs (*Figure 7D–E*). We found no differences in the control group (*Figure 7H–K*).

We also quantified the percentage of PyCs that significantly altered their overall activity level upon ChC silencing. Significantly more neurons increased their activity after SalB injection than in the control group (KORD: 21%, control: 13%, $\chi^2$-test p<0.001), while in contrast fewer neurons decreased their activity (KORD: 7%, control: 11%, $\chi^2$-test p<0.001). Finally, to test whether the effect of silencing ChCs preferentially affected PyCs that were weakly or strongly modulated by locomotion, we correlated the locomotion modulation index (LMI) for each PyC with the change in $\Delta F/F$ due to ChC silencing (after minus before). The correlation was weak and not significant ($r=0.0068$, p=0.7827), showing that ChCs do not differentially target PyCs based on locomotion modulation.

In contrast to the idea that ChCs exert powerful control over AP generation, these results suggest that ChCs only weakly modulate PyC activity, affecting only a relatively small population of cells.

## Discussion

In this study we investigated the connectivity and function of ChCs in V1. We find that ChCs receive input from local L5 PyCs and higher-order cortical regions and show strong arousal-related activity. While they are also visually responsive, ChCs are weakly selective for stimulus content but respond strongly to visuomotor mismatch events. Furthermore, we found that repeated exposure to a virtual tunnel induces plasticity of the visual responses of ChCs as well as their axo-axonic synapses at L2/3 PyC AISs. Finally, ChCs only weakly inhibit PyCs.

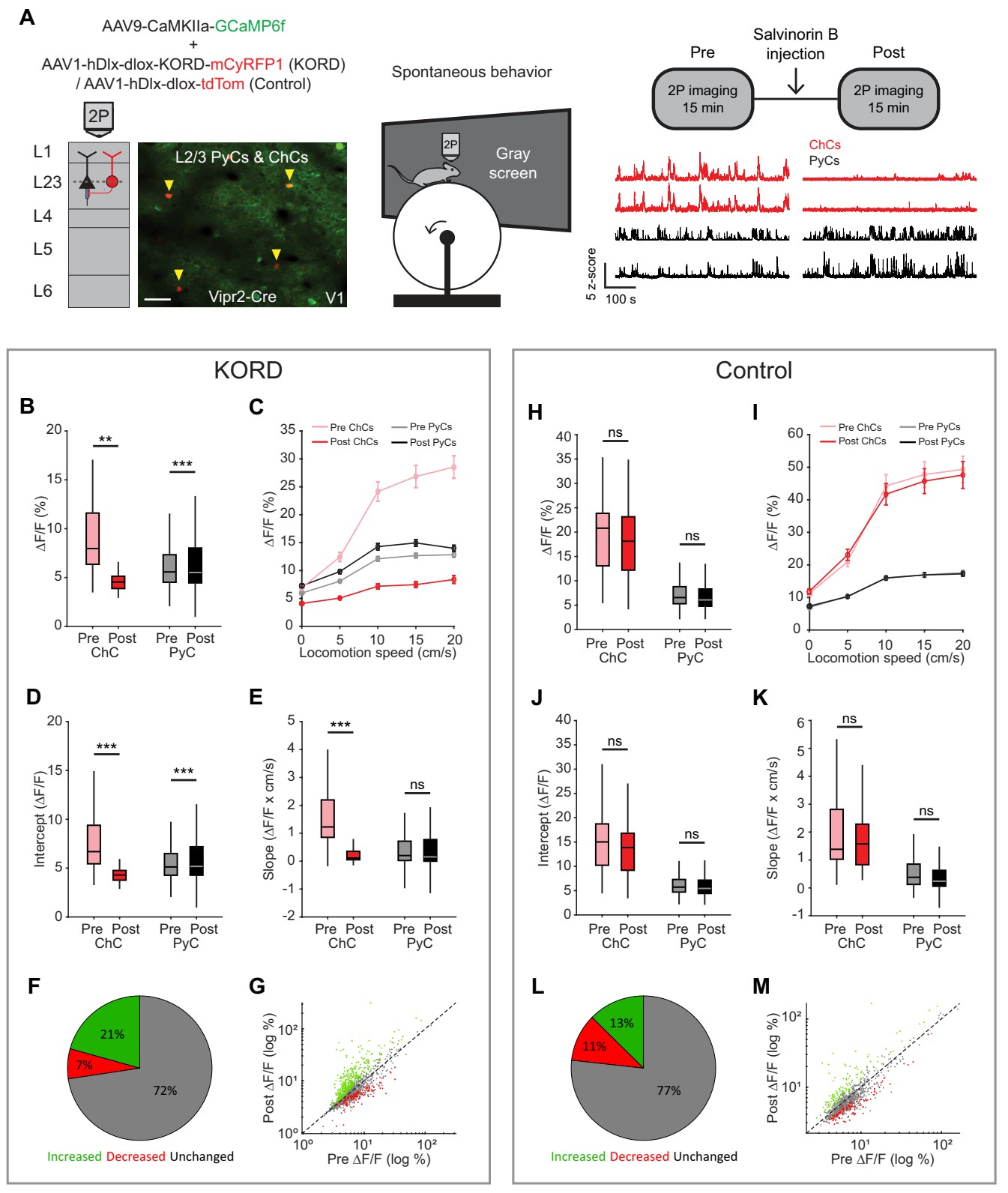

**Figure 7.** Chandelier cells (ChCs) weakly inhibit pyramidal cell (PyC) activity independent of locomotion speed. (**A**) Viral strategy and experimental design for chemogenetic silencing of ChCs (yellow arrows) in awake mice. Activity of ChCs and PyCs was recorded before and after an s.c. injection of Salvinorin B (SalB) (10 mg/kg). Bottom right: example traces of two ChCs and two L2/3 PyCs before and after injection of SalB. Scale bar example field of view, 50 μm. (**B–G**) Activity and locomotion modulation fit parameters in mice expressing KORD-tdTom in ChCs. (**B**) Average ChC activity decreases,

*Figure 7 continued on next page*

*Figure 7 continued*

while PyC activity increases after injection of SalB. LMEM for all comparisons, \*\*\*: p<0.001, \*\*: p<0.01, \*: p<0.05, ns: not significant (*n*=84 ChCs and 1669 PyCs, 12 sessions from 12 mice). Box plots represent median, quantiles, and 95% confidence interval (CI) over neurons. (C) Activity of ChCs and PyCs as a function of locomotion speed before and after injection of SalB. Points and error bars represent mean ± SEM, respectively. (D) Average intercept of locomotion modulation fit before and after injection of SalB. The intercept decreases for ChCs but increases for PyCs. (E) Average slope of locomotion modulation fits before and after injection of SalB. The slope decreases for ChCs but remains unchanged in PyCs. (F) Pie chart with the percentage of neurons that showed significantly increased (green), significantly decreased (red), or unchanged (gray) activity after SalB injection. (G) Scatter plot showing for each neuron its activity pre and post SalB injection. (H–M) Activity and locomotion modulation fit parameters in mice expressing tdTom in ChCs (control group). (H) As in B, but for the control group (*n*=37 ChCs and 968 PyCs, 7 sessions from 7 mice). (I) As in C, but for control group. (J) As in D, but for the control group. (K) As in E, but for the control group. (L) As in F, but for the control group. (M) As in G, but for the control group.

Recent EM reconstructions have shown that although most L2/3 PyCs in V1 receive ChC input, the number of ChC boutons on the AISs of PyCs varies significantly and correlates with their size and laminar depth (*Schneider-Mizell et al., 2021*). Accordingly, we found that ChCs provided GABAergic inputs to many of the L2/3 PyCs in their vicinity. Unexpectedly, our retrograde rabies virus tracing experiments showed that ChCs receive most of their input from local L5 PyCs. In keeping with the identified circuits, we did not detect L2/3 PyCs providing inputs to ChCs using dual-patch clamp recordings, suggesting that the rabies labeled neurons in layers 1–4 were predominantly interneurons, in line with previous work (*Jiang et al., 2015*). ChCs also received long-range inputs from various thalamic and cortical regions (e.g. dLGN, LP, RSC, S1), matching those innervating other interneuron subsets in V1 (*Ma et al., 2021*). Optogenetic activation of cortical feedback induced responses in almost all V1 ChCs and repeated stimulation caused synaptic depression, indicating a high release probability, which is usually seen in strong cortical inputs to V1 interneurons (*Ma et al., 2021*). Taken together, the connectivity pattern of ChCs in V1 thus appears to be non-reciprocal, as is also observed in prelimbic cortex (*Lu et al., 2017*). However, in contrast to prelimbic cortex where ChCs receive input from deep layer 3 PyCs, we show here that V1 ChCs receive most of their input from layer 5.

Although ChCs in V1 are sparse and the number of starter ChCs in our rabies tracing experiments was relatively low, we found large numbers of L5 PyCs providing monosynaptic input to them. This means that each ChC receives input from many L5 PyCs. This convergence of L5 PyC inputs, if not strongly organized, could explain the low selectivity of ChC responses we observed to natural images compared to those of L2/3 and L5 PyCs. This is also supported by the MEIs we obtained from the pre-trained CNN. MEIs of L5 PyCs were often high-contrast oriented patterns, like those of L2/3 PyCs, or more complex textures. Instead, MEIs of ChCs were of lower contrast and less structured and similar to MEIs created to maximally drive large numbers of L5 PyCs. Combined inputs from L5 PyCs with diverse tuning properties are also expected to result in tuning to low SFs, large RFs, and low orientation tuning, matching the modeled ChCs. Finally, modeled ChCs also displayed weak contrast tuning. This is an expected consequence of the preference for low SFs, as these require lower contrast to be detected (*Heimel et al., 2010*; *Boynton et al., 1999*).

In our virtual tunnel paradigm, ChCs were active during locomotion but even more so during visuomotor mismatch caused by halting the tunnel while the mice were running. Surprisingly, ChC activity was suppressed by the visual stimuli in the tunnel, or when the visual flow started while the mice were stationary. We found that this apparent discrepancy with the strong responses to natural stimuli could be explained by experience-dependent plasticity of ChC activity: while ChCs responded vigorously to oriented-bar stimuli in naive mice, these responses were almost absent after training in the virtual tunnel. While the underlying plasticity mechanisms remain to be investigated, there are several possibilities. First, it could be caused by reduced excitatory synaptic transmission of visual input to ChCs or increased visually driven inhibition. Second, the plasticity could involve changes in the response properties of L5 PyCs providing input to ChCs. Interestingly, recent evidence shows that during visuomotor behavior, intratelencephalic (IT) L5 PyCs respond in a similar fashion as ChCs in that they respond to running and visuomotor mismatch but are suppressed by unexpected visual flow (*Heindorf and Keller, 2022*). Whether L5 IT PyCs are the main L5 PyC subset providing input to ChCs remains to be investigated. A third option is that the experience-dependent changes in ChC activity are related to arousal. ChCs were previously shown to display arousal-related activity in various brain regions (*Schneider-Mizell et al., 2021*; *Dudok et al., 2021*; *Bugeon et al., 2022*; *Bienvenu et al., 2012*; *Massi et al., 2012*), which matches our observation that their activity is highly correlated with

running and pupil size. In addition, the neuromodulator acetylcholine is involved in axonal arborization of developing ChCs, further underlining the relationship between ChCs and arousal (*Steinecke et al., 2022*). During training in the virtual tunnel, mice become acquainted with visual stimuli and learn that running consistently results in visual flow. After training, visual stimuli may thus cause less arousal while unexpected visual flow halt during running may cause more. In line with this idea, it was recently shown that visuomotor mismatch induces strong noradrenergic input to V1 (*Jordan and Keller, 2023*). Noradrenergic input has been shown to depolarize ChCs in frontal cortex (*Kawaguchi and Shindou, 1998*). However, in contrast to experiments performed in prelimbic cortex (*Lu et al., 2017*), we did not identify neuromodulatory inputs to ChCs in our rabies tracing experiments. Possibly, these inputs act predominantly through extrasynaptic receptors and are less efficiently labeled by the transsynaptic rabies approach. As ChCs appear to have a higher prevalence in prelimbic cortex (*Lu et al., 2017*), transsynaptic labeling may be more effective here than in V1. Future experiments may reveal to what extent plasticity of ChC responses is stimulus specific or arousal related.

When examining how neurons respond to expected or unexpected visual input, it is useful to consider the concept of predictive coding (*Friston, 2012*; *Rao and Ballard, 1999*; *Shipp, 2016*; *Summerfield et al., 2011*; *den Ouden et al., 2012*; *Lee and Mumford, 2003*; *Keller and Mrsic-Flogel, 2018*; *Egner and Summerfield, 2013*). In this framework, an internal representation of the world is compared with sensory inputs and updated based on prediction errors: the differences between the expected and actual sensory inputs. These prediction errors can be negative, signaling the absence of an expected visual stimulus, or positive, signaling the presence of an unexpected visual stimulus. It has been suggested that prediction errors may be computed by L2/3 PyCs, while the internal representation is encoded by local L5 PyCs and top-down inputs from higher-order brain regions (*Heindorf and Keller, 2022*; *Keller and Mrsic-Flogel, 2018*; *Bastos et al., 2012*). Previous work has shown that in V1, L2/3 PyCs indeed respond to mismatches between visual flow and locomotion (*Attinger et al., 2017*; *Jordan and Keller, 2020*; *Keller et al., 2012*; *Zmarz and Keller, 2016*). Recent evidence suggests that two genetically defined subpopulations of L2/3 PyCs preferentially respond to (unexpected) visual stimuli, which could be considered positive prediction errors, or visuomotor mismatch, which could be seen as negative prediction errors (*O'Toole et al., 2023*). In line with these findings, we observed two functional clusters of PyCs in L2/3, one responding predominantly to visual stimuli (V-PyCs) and the other to visuomotor mismatch (NV-PyCs).

Prediction error responses require inhibitory neurons to compute the difference between expectation and actual visual input. As ChCs receive input from L5 and higher-order cortical regions, they might contribute to calculating positive prediction errors by subtracting the internal representation (i.e. the predicted stimulus) from the actual visual input. However, as ChCs only mildly inhibited L2/3 PyCs, they may not be well suited for this function. Moreover, given that we and others (*Schneider-Mizell et al., 2021*; *Dudok et al., 2021*; *Jung et al., 2023*) have found that ChCs are highly correlated in their activity, it seems unlikely that they contain high-dimensional prediction error information. Predictive coding also involves updating of the internal representation based on prediction errors. Recent evidence suggests that this plasticity requires a gating signal that is provided by noradrenergic input (*Jordan and Keller, 2023*). One option is therefore that ChCs also encode arousal and play a role in the regulation of plasticity.

An intriguing possible mechanism through which ChCs could regulate plasticity is by controlling geometric modifications of the AIS of L2/3 PyCs. We found that visuomotor experience in the virtual tunnel resulted in AIS shortening and more AISs with putative ChC contacts. This is in agreement with previous studies revealing modifications in the density distribution of ChC boutons on PyC AISs in premotor cortex upon motor learning, and in S1 upon chemogenetically activating L2/3 PyCs (*Pan-Vazquez et al., 2020*; *Jung et al., 2023*). Changes in ChC innervation may alter the excitability of PyCs by reducing AP generation. Considering our observation that in vivo chemogenetic silencing of ChCs only mildly increased L2/3 PyC activity, their electrical contribution to vetoing AP output seems, if anything, limited and affects only a small proportion of cells. These findings are thus in contrast with the general notion that ChCs exert powerful control over PyC output (*Jung et al., 2022*; *Gallo et al., 2020*), but consistent with computational simulations predicting a relatively small inhibitory effect of GABAergic innervation of the AIS, possibly involving shunting inhibition (*Douglas and Martin, 1990*; *Shang et al., 2023*). One explanation for the weak effects we observed is the high variability in the number of GABAergic boutons that PyCs receive at their AISs. Possibly, only a smaller fraction

of PyCs with high numbers of AIS synapses are inhibited when ChCs are active. Indeed, we find that only a small fraction of PyCs increased their activity upon chemogenetic silencing of ChCs, in line with findings by others showing that manipulating ChC activity in vivo has relatively weak effects on small populations of PyCs (*Lu et al., 2017*; *Dudok et al., 2021*).

Increasing L2/3 PyC activity, like visuomotor experience, also decreased AIS length (*Jamann et al., 2021*), which is known to reduce PyC excitability (*Jamann et al., 2021*) and likely represents a homeostatic plasticity mechanism. Whether such geometric AIS plasticity is regulated by ChCs is not known. Interestingly, ChC boutons face postsynaptic sites where cisternal organelles are located with micrometer precision (*Schneider-Mizell et al., 2021*; *Benedeczky et al., 1994*). As cisternal organelles are implicated in calcium signaling and AIS plasticity (*Schlüter et al., 2017*), ChCs may thus effectively influence these events. In this scenario, ChC activation could prevent homeostatic AIS shortening of L2/3 PyCs if their activity occurs during behaviorally relevant, arousal-inducing events. Testing this hypothesis would require, for example, studying the consequences of synchronous or asynchronous activation of ChCs and PyCs on AIS plasticity. The emerging availability of mouse models to genetically target ChCs (*Schneider-Mizell et al., 2021*; *Tasic et al., 2018*; *Lu et al., 2017*; *Dudok et al., 2021*; *Daigle et al., 2018*) and novel live markers of the AIS (*Thome et al., 2023*) are making such experiments possible.

In conclusion, our comprehensive study of the function and connectivity of ChCs in V1 reveals that ChC activity is primarily driven by arousal-inducing events such as unexpected visual stimuli, visuomotor mismatch, and locomotion. We also observed remarkable plasticity of ChC responses by visuomotor experience, as well as in their innervation of the AIS. Interestingly, our results indicate that in vivo, ChCs do not exert strong control over AP generation at the AIS, but only provide a weak inhibitory influence on PyC activity. Future experiments may reveal whether ChCs provide a gating signal for AIS plasticity and elucidate the underlying mechanisms.

## Materials and methods

**Key resources table**

| Reagent type (species) or resource | Designation | Source or reference | Identifiers | Additional information |
|---|---|---|---|---|
| Strain, strain background (*Mus musculus*, male and female) | Vipr2-Cre; C57BL/6J | Jackson Laboratories | 031332 | |
| Strain, strain background (*Mus musculus*, male and female) | Pvalb-T2A-FLpO-D; C57BL/6J | Jackson Laboratories | 022730 | |
| Strain, strain background (*Mus, musculus*, male and female) | AI65(RCFL-tdT)-D; C57BL/6J | Jackson Laboratories | 021875 | |
| Strain, strain background (*Mus musculus*, male and female) | Rbp4-Cre; C57BL/6J | GENSAT project | KL100 | |
| Strain, strain background (*Mus musculus*, male and female) | CBA/JRj | Janvier Labs | | |
| Transfected construct (adeno-associated virus) | AAV9-CaMKIIa-GCaMP6f | Addgene | 100834-AAV9 | |
| Transfected construct (adeno-associated virus) | AAV1-CAG-flex-mRuby2-GCaMP6f | Addgene | 6719-AAV1 | |
| Transfected construct (adeno-associated virus) | AAV1-hSyn1-GCaMP6f | Addgene | 100837-AAV1 | |
| Transfected construct (adeno-associated virus) | AAV9-CaMKIIa-jGCaMP8m | VVF Zurich | v630-9 | |
| Transfected construct (adeno-associated virus) | AAV-PHP.eB-shortCAG-dlox-GCaMP6f(rev)-dlox | VVF Zurich | v657-PHP.eB | |
| Transfected construct (adeno-associated virus) | AAV5-hDlx-DIO-eYFP-t2A-TVA | Dr. Seungho Lee | | |
| Transfected construct (adeno-associated virus) | AAV9-hDlx-DIO-oG | Dr. Seungho Lee | | |

*Continued on next page*

*Continued*

| Reagent type (species) or resource | Designation | Source or reference | Identifiers | Additional information |
|---|---|---|---|---|
| Transfected construct (rabies virus) | Rbv-ΔG-mCherry | Charite Berlin | BRABV-001 | |
| Transfected construct (adeno-associated virus) | AAV1-hDlx-dlox-hKORD-mCyRFP1(rev)-dlox | VVF Zurich | v326-1 | |
| Transfected construct (adeno-associated virus) | AAV1-hDlx-dlox-mCyRFP1(rev)-dlox | VVF Zurich | v313-1 | |
| Transfected construct (adeno-associated virus) | AAV9-CaMKIIa-ChR2-eYFP | Addgene | 26969-AAV9 | |
| Transfected construct (adeno-associated virus) | AAV1-hDlx-dlox-ChrimsonR-tdTomato(rev)-dlox | VVF Zurich | v674-1 | |

## Mice

All experiments were approved by the institutional animal care and use committee of the Royal Netherlands Academy of Arts and Sciences under Central Committee Animal experiments (CCD) licenses AVD 80100 2017 1045, AVD 80100 2022 15934, and AVD 80100 2022 15935, Academy of Arts and Sciences. We used both male and female mice for all experiments. For a subset of tunnel experiments, we used Vipr2-Cre mice crossed with Pvalb-T2A-FlpO-D and AI65(RCFL-tdT)-D (Jackson Laboratories, https://www.jaxmice.jax.org/, strain 031332, 022730, and 021875, respectively, *Schneider-Mizell et al., 2021*; *Daigle et al., 2018*). For experiments with natural images (*Figure 3*), we used Vipr2-Cre mice crossed with CBA/JRj (Janvier labs) mice for targeting ChCs and L2/3 PyCs and Rbp4-Cre mice (line KL100, GENSAT project) for targeting L5 PyCs. We used Vipr2-Cre mice for all other experiments. Mice were group housed under a 12 hr reversed day/night cycle and provided with ad libitum access to food. Mice were water-deprived during the training and imaging phase for the virtual tunnel task. Before and after the training phase mice had ad libitum access to water. Experiments were performed in the dark phase. Mice had access to a running wheel in their home cage throughout the duration of the experiment.

## Viral injections and window surgery

For all cranial surgical procedures, mice were anesthetized with isoflurane (4% induction, 1.6% maintenance in oxygen). During the surgeries, temperature was maintained at 37 degrees with euthermic pads and eyes were protected from light and drying using Cavasan eye ointment. For viral injections targeting L2/3 in all experiments except those used for CNN modeling, the skull was exposed, three small craniotomies were drilled in the skull overlying right V1 (centered around 2.9 mm lateral, 0.5 mm anterior to lambda), and one injection of 70–120 nl virus (titer ~10E12 viral genomes per ml) in each craniotomy at a depth of approximately 250 μm was made. We used one or more of the following viruses as indicated in the figures: AAV9-CaMKIIa-GcaMP6f, AAV1-CAG-flex-mRuby2-GcaMP6f, AAV1-hSyn1-GcaMP6f, AAV-PHP.eB-shortCAG-dlox-GcaMP6f(rev)-dlox, AAV5-hDlx-DIO-eYFP-t2A-TVA, AAV9-hDlx-DIO-oG, Rbv-ΔG-mCherry, AAV1-hDlx-dlox-hKORD-mCyRFP1(rev)-dlox, AAV1-hDlx-dlox-ChrimsonR-tdTomato(rev)-dlox, AAV9-CaMKIIa-ChR2-eYFP (also see Key resources table). For L2/3 CNN model experiments (*Figure 3*) we drilled 7 holes across the visual cortex (centered around 2.9 mm lateral, 0.5 mm anterior to lambda) and injected 70–120 nl virus (AAV9-CaMKIIa-GcaMP6f and AAV1-hDlx-dlox-mCyRFP1(rev)-dlox, titer ~0.7 × 10E12 viral genomes per ml) across two depths (200 and 400 μm). For L5 CNN model experiments (*Figure 3*), we performed i.v. injections in the tail vein of awake body-restrained mice using 125 μl virus (AAV-PHP.eB-shortCAG-dlox-GcaMP6f(rev)-dlox, titer ~3.4 × 10E12 viral genomes per ml). All mice were allowed to recover from viral injections for 2 weeks.

For the cranial window surgery, mice were implanted with a double (3+4 mm diameter) or triple (4+4+5 mm diameter, for L2/3 CNN experiments) glass window on the center of V1. To allow head fixation, mice were implanted with a custom metal head ring. The glass window and head ring were fixed to the skull using dental cement. At least 1 week after surgery, mice were handled and habituated

to being head immobilized in our setup until they sat comfortably and started running regularly on their own (typically ~5 sessions).

## Visual stimuli and virtual tunnel

All visual stimuli for two-photon imaging were presented on a gamma-corrected full HD LED monitor using OpenGL and Psychophysics Toolbox 3 running on MATLAB (Mathworks). The monitor was positioned 15 cm from the mouse. For passive visual stimulation before and after training, we used full screen square gratings (0.05 cpd) of different contrasts (0.05–0.1–0.2–0.4–0.6–0.8–1) moving in one of eight different directions (45 degrees apart, 1 cps). Stimuli were presented for 1 s with a random inter-trial interval of 4–6 s and repeated a minimum of eight times each.

To model visual RF selectivity using the CNN model, we used 3600 unique images from 720 classes taken from the THINGS database (*Hebart et al., 2019*). In order to test reliability across repetitions of the same stimuli and model cross-validated correlations, an additional 40 unique images were repeated 10 times. Individual images were composed of two square images from the same category, blended to cover the whole screen. Images were shown for 0.5 s, followed by a delay of 0.5 s where mice viewed a gray screen. CNN experiments were performed on naive mice.

For our virtual reality setup, we measured absolute running speed via a rotary encoder, which enabled real-time rendering of the virtual corridor. The left half of the corridor was displayed on a monitor positioned at a 45 degree angle and viewed through a mirror, giving the perception of a symmetrical tunnel. The virtual environment was created using Psychophysics Toolbox 3 and OpenGL running on MATLAB. The corridor was 100 cm long and its walls were covered in a black and white Gaussian noise texture, with visual stimuli superimposed. This included three vertical gratings and three checkerboard stimuli placed 11 cm apart between the distances of 22 and 77 cm (*Figure 4B*). Following a run through the visual part of the corridor, mice were exposed to a luminance-matched gray screen, followed by an 8 kHz auditory cue after 1 s, and then received a 5 µl water reward 2 s later. Mice were trained over a minimum of six sessions, with one session per day, until they consistently completed over 70 trials. Mice continued to be trained in-between imaging sessions until the experiment was completed. For open loop experiments, visual flow speed was set at a constant 20 cm/s. Visuomotor mismatches were only introduced during the experimental sessions and were achieved by briefly halting the visual flow for 0.5 s at random locations in the visual section of the tunnel where visual stimuli appeared (between 20 and 80 cm). Mismatches were restricted to periods where the mouse was running during closed loop sessions and periods where the mouse was stationary during open loop sessions.

## Two-photon calcium imaging

### Data collection

Two-photon imaging experiments were performed on a two-photon microscope (Neurolabware) equipped with a Ti-sapphire laser (Mai-Tai 'Deepsee', Spectraphysics; wavelength, 920 nm) and a 16×, 0.8 NA water immersion objective (Nikon) at a zoom of 1.6× for L2/3 imaging and a zoom of 2× for L5 imaging. The microscope was controlled by Scanbox (Neurolabware) running on MATLAB. Images were acquired at a frame rate of 15.5 or 31 Hz. In some sessions, we performed dual-plane imaging at 31 Hz using an electrically tunable lens (OptoTune), resulting in an effective frame rate of 15.5 Hz per plane. For L2/3 experiments, somatic imaging was performed at ~150–300 µm depth. For L5 experiments, we recorded apical dendrites at a depth of ~200 µm. Pupil size and position was tracked at the imaging frame rate using an IR camera (Dalsa Genie).

### Preprocessing

We used the SpecSeg toolbox for preprocessing as described in detail before (*de Kraker et al., 2022*). In short, we performed rigid motion correction using NoRMCorre (*Pnevmatikakis and Giovannucci, 2017*) followed by automated region-of-interest (ROI) selection based on cross-spectral power across pixels. After manual refinement, raw ROI signals were extracted and corrected for neuropil as described before (*de Kraker et al., 2022*) by subtracting the average pixel values in an area surrounding each ROI multiplied by 0.7. $\Delta F/F$ traces were calculated as $\Delta F/F = (F - F0)/F0$, in which $F0$ represents a moving baseline (10th percentile over 5000 frame windows). Matching of chronically recorded neurons was performed using the chronic matching module of SpecSeg.

## Passive sessions

For comparing single-cell responses between conditions we first matched neurons that we chronically imaged across multiple sessions. For subsequent analyses we only selected neurons that we found back in all sessions relevant for that analysis (e.g. pre-post or pre-tunnel-post). For statistical analyses, we took the mean $\Delta F/F$ trace over all stimuli and computed the baseline-corrected average from 0.2 to 1.2 s after stimulus onset for each cell. Baseline correction (here and below) was performed by subtracting the average $\Delta F/F$ before stimulus onset ($t<0$ s) from the average trace. We removed all trials in which the running speed exceeded 1 cm/s anywhere from 1 s before until 4 s after stimulus onset.

To correlate activity with estimates of arousal, mice were allowed to freely run or rest in front of a gray monitor for ~10 min. We computed the correlation coefficient between calcium activity of each cell with running speed and pupil size using the 'corr' function in MATLAB. For pairwise correlations between neurons within each cell type, we computed the average correlation coefficient during stationary periods (running speed <1 cm/s) for each cell with all other cells of that type within the field of view. For displaying purposes (*Figure 2B*), we z-scored single-cell traces over the entire 10 min session.

For orientation and direction tuning we took the baseline-corrected average from 0.2 to 1.2 s after stimulus onset for each cell. For tuning calculations in *Figure 2* and *Figure 5*, we only included contrasts higher than 0.7. For orientation tuning, the mean responses to the eight orientations ($\theta$) were fit with a single circular Gaussian using nonlinear least-squares fitting as follows:

$$R\left(\theta\right) = C + R_p e^{\frac{-ang_{ori}\left(\theta - \theta_{pref}\right)^2}{2\sigma^2}} \tag{1}$$

where $R(\theta)$ is the response to the grating of orientation $\theta$, $C$ is an offset term, $R_p$ is the response to the preferred orientation, $ang_{ori}(x)=min[abs(x),\ abs(x-180),abs(x+180)]$ wraps angular differences to the interval 0 to 90 degrees, and $\sigma$ is the standard deviation of the Gaussian. For direction tuning, the mean responses to the eight directions ($\theta$) were fit with a double circular Gaussian:

$$R\left(\theta\right) = C + R_p e^{\frac{-ang_{dir}\left(\theta - \theta_{pref}\right)^2}{2\sigma^2}} + R_n e^{\frac{-ang_{dir}\left(180 + \theta - \theta_{pref}\right)^2}{2\sigma^2}} \tag{2}$$

where $R_p$ is the fitted response at the preferred direction, $R_n$ is the fitted response at the non-preferred direction, $ang_{dir}(x)=min[abs(x),\ abs(x-360),abs(x+360)]$ wraps angular differences to the interval 0 to 180 degrees.

Orientation tuning strength was calculated using 1-CircVar:

$$1 - CircVar = \left|\frac{\sum_k R\left(\theta_k\right) e^{2i\theta_k}}{\sum_k abs\left[R\left(\theta_k\right)\right]}\right| \tag{3}$$

where $R(\theta_k)$ is the response to the orientation $\theta_k$ (in radians). Direction tuning strength was also assessed using 1-CircVar using the following equation for directional data:

$$1 - CircVar = \left|\frac{\sum_k R\left(\theta_k\right) e^{i\theta_k}}{\sum_k abs\left[R\left(\theta_k\right)\right]}\right| \tag{4}$$

where $\theta_k$ is the direction of the grating (in radians). Orientation and direction tuning responses were normalized to their maximum response before calculating 1-CircVar. Orientation and direction tuning curves used for plotting were computed by shifting the average tuning curve to the cell's fitted preferred direction, followed by averaging across cells to compute cell type averages.

## CNN model

As input to our CNN model we used spike probabilities generated using CASCADE (*Rupprecht et al., 2021*) from neuronal responses to 3600 natural images. We took the single-cell average spike probability between 0 and 0.5 s after stimulus onset as the response to each natural image. To model the neuronal responses we employed a pre-trained CNN (*Szegedy et al., 2014*) ('Inception v1') and fit a

mapping function to the activations of a target layer (*Bashivan et al., 2019*). In brief, for each neuron we learned a set of 2D spatial weights ($W_s$) with size equal to the size of a channel of the target CNN layer (i.e. the pixels) and a set of feature weights ($W_f$) with size equal to the number of channels of the target CNN layer. The $W_s$ learned the spatial RF of each neuron, while the $W_f$ learned their feature selectivity as a weighted sum over the features of the pre-trained CNN layer. Thus, the predicted response of a neuron $n$ (i.e. $\widehat{y_n}$) was computed according to:

$$\widehat{y_n} = \left[ \sum W_s^{(n)} * X_l \right] * W_f^{(n)} \tag{5}$$

where $X_l$ are the batch-normalized activations from the target CNN layer $l$. The weights were jointly optimized across all neurons to minimize the prediction error $\mathcal{L}_{error}$ regularized by a combination of a smoothing Laplacian loss ($\mathcal{L}_{Laplace}$: see below) over $W_s$ (*Bashivan et al., 2019*), to encourage smooth spatial RFs, a L1 loss ($\mathcal{L}_1$: see below) over $W_f$ to encourage sparsity/selectivity of the features (*Cadena et al., 2019*; *Walker et al., 2019*) and a L2 loss ($\mathcal{L}_2$: see below) over both parameters (*Walker et al., 2019*; *Bashivan et al., 2019*). Thus the cost function was computed as follows:

$$\mathcal{L}_{error} = \sqrt{\sum_n \left( \widehat{y}_n - y_n \right)^2} \tag{6}$$

$$\mathcal{L}_{Laplace} = \lambda_s \sqrt{\sum_n \left( W_s^{(n)} * L \right)^2}, \ L = \begin{bmatrix} 0 & -1 & 0 \\ -1 & 4 & -1 \\ 0 & -1 & 0 \end{bmatrix} \tag{7}$$

$$\mathcal{L}_1 = \lambda_f \sum_n \left| W_f^{(n)} \right| \tag{8}$$

$$\mathcal{L}_2 = \lambda_2 \sum_n (W_s^{(n)^2} + W_f^{(n)^2}) \tag{9}$$

$$\mathcal{L} = \mathcal{L}_{error} + \mathcal{L}_{Laplace} + \mathcal{L}_1 + \mathcal{L}_2 \tag{10}$$

where $y_n$ is the response of neuron $n$. The hyper-parameters ($l$, $\lambda_s$, $\lambda_f$, $\lambda_2$ and the learning rate) were selected using a grid search and then selecting the combination yielding the highest correlation in the training set. For all cell types, the best target CNN layer ($l$) was 'conv2d2'. The model was implemented in PyTorch (*Paszke et al., 2021*) and optimized with Adam (*Kingma and Ba, 2017*) in 600 epochs with a batch size of 128 and early stopping (every 200 epochs, decay factor = 3).

After training, we generated the MEIs by optimizing the pixels in the input to maximize the response of each model neuron independently. To ensure the stability and interpretability of the MEIs, we introduced preconditioning and regularizations to avoid high-frequency artifacts (*Olah et al., 2017*), and specifically frequency penalization and transformations (padding, jittering, rotation, and scaling). Optimization of the MEIs was done using Lucent (i.e. PyTorch implementation of Lucid, *Olah et al., 2017*) with Adam (learning rate = 1e-2, weight decay = 1e-3) in 50 epochs. To generate MEIs for combinations of L5 PyCs (*Figure 3—figure supplement 1F*), we first randomly chose 10 starter L5 PyCs. On each iteration we then added five random L5 PyCs (except for the first iteration, where we added four) and optimized a 'composite MEI' by maximizing the summed responses from all neurons.

Furthermore, for each neuron we computed a few metrics either directly on the data or by analyzing model responses to novel stimuli. The response strength of each neuron was computed by taking the maximum response of the time course averaged over all images, divided by the standard deviation of the response in the 200 ms before the stimulus onset. In addition to the 3600 images used for training, we also recorded responses to 40 images that were presented 10 times each (4000 images in total) to test the generalization performance of the model. We computed the selectivity to visual input as the sparsity of their response distribution (*Zoccolan et al., 2007*), going from 0 (not sparse) to 1 (maximally sparse) as follows:

$$\text{Sparsity} = [1 - \frac{\sum \left( \frac{R_i}{N} \right)^2}{\left( \frac{\sum R_i^2}{N} \right)}]/[1 - \left( \frac{1}{N} \right)] \tag{11}$$

where $R_i$ is the response to $i$th image and $N$ is the number of test images (40 here).

For population decoding, we performed LDA using the 'fitcdiscr' function in MATLAB on the neuronal responses to the 40 test images. For each decoding run, we trained an LDA decoder by randomly selecting 80% of the trials and then testing the decoder on the remaining 20% of trials. We repeated this procedure 50 times to account for variability between trials. For ChCs, we used the total population ($n$=34) and averaged the resulting 50 decoding accuracies. For L2/3 PyCs, we performed 1000 iterations of 50 runs each by randomly subsampling 34 neurons on each iteration. For each iteration we then averaged the 50 runs, resulting in a distribution of 1000 decoding accuracies. Statistical significance was determined using a permutation test of the average ChC decoding accuracy (12.55%) versus the distribution of 1000 L2/3 PyC accuracies. Chance level decoding was at 2.5% (40 images total).

We also computed the oracle correlation of each neuron by correlating the responses to multiple repetitions of the same 40 test images and then taking the average correlation (*Walker et al., 2019*). Only neurons with an oracle correlation higher than zero were included in the model. In order to test the performance of the model we correlated the predicted responses with the recorded neuronal responses to the 40 test images (not used for training). The resulting 'cross-validation correlations' were largely identical to the oracle correlations for all cell types (*Figure 3—figure supplement 1B–C*).

We computed the RF size as the full width at half maximum of a 2D Gaussian fitted to $W_s$ of each neuron. Neurons with RF size >40 degrees were excluded from analysis.

We determined the orientation tuning by showing to the model neurons full-contrast Gabor gratings with 8 different orientations (22.5 degrees apart), 5 SFs (from 0.02 to 0.08 cycles per degree), and 2 phases (0 and 90 degrees). The OSI was computed using 1-CircVar as above.

For SF tuning we used 12 SFs (0.01, 0.02, 0.03, 0.04, 0.05, 0.06, 0.08, 0.1, 0.2, 0.3, 0.4, 0.6). The preferred SF was defined as the SF at which the max response for each neuron occurred.

We measured contrast sensitivity using full-screen gratings of increasing contrast (0.05, 0.1, 0.2, 0.4, 0.6, 0.8, 1) followed by calculating a linear fit across the contrast tuning responses. We took the slope of the fit as the contrast sensitivity for each neuron.

For pairwise correlations of spike probabilities between neurons within each cell type, we computed the average correlation coefficient during stationary periods (running speed <1 cm/s) for each cell with all other cells of that type within the field of view.

## Virtual tunnel

For the analysis of tunnel data, we removed trials that took longer than 15 s from start until reward delivery. To compute activity as a function of location in the tunnel (0–100 cm) we divided the tunnel into 50 bins of 2 cm each and averaged activity of all frames occurring in a bin, followed by averaging over trials for each cell. Activity for each cell in the non-visual part of the tunnel was time-locked to exit from the visual part until 9 s later and subsequently averaged over trials. For this analysis we removed all trials containing visuomotor mismatch events. Locomotion traces were made by averaging locomotion speed across all trials for each session, followed by averaging normalized traces (between 0 and 1) over mice. For single-cell average activity in the visual part of the tunnel we took the mean activity between 20 and 80 cm and subtracted the mean activity between 0 and 20 cm. For the non-visual part, we averaged the entire trace for each cell.

For clustering, we z-scored trial-averaged traces in the visual section of the tunnel from all PyCs as input for the 'linkage' function in MATLAB with 'ward' as method. We then performed clustering using the 'cluster' function with the optimal number of clusters as determined by silhouette evaluation of the traces. Since the result of silhouette evaluation could not exclude the possibility that only one cluster was optimal, we tested the separability of the V-PyCs and NV-PyCs clusters using a permutation test. In order to do this, we first computed an average trace combined for the visual and non-visual section for each cell. We then z-scored this trace, and computed the visual response by averaging the z-scored values between 20 and 80 cm in the visual section of the tunnel. This procedure therefore gave us a single visual responsiveness value for each cell. Next, we binned this distribution of visual responsiveness over neurons into $n$ bins, and normalized the sum of all counts to 1, such that $P$ and $Q$ describe the discretized probability mass functions for the V-PyCs and NV-PyCs respectively. We then calculated the separability in visual responsiveness between the V-PyCs and NV-PyCs clusters using the Bhattacharyya distance (BD):

$$BD = -\ln\left(\sum_{i=1}^{n}\sqrt{P_i Q_i}\right) \tag{12}$$

Here, $P_i$ and $Q_i$ are the frequencies of samples in 'visual response' bin $i$. The real data we observed gave a BD of 0.4262. Finally, we quantified whether this separability value was higher than could be expected by chance, and therefore whether the V-PyCs and NV-PyCs groups indeed represented distinct clusters. To this end, we generated a distribution of 1000 randomized BD values obtained by shuffling the visual and non-visual responses of all cells, and repeating the procedure as described above. Statistical significance was determined using a permutation test of 0.4262 versus the distribution of 1000 shuffle-randomized populations.

We calculated mismatch responses after correcting for location of the mismatch events, as spatial location in the tunnel can modulate activity of V1 cells even when visual input is identical (*Saleem et al., 2018*; *Fiser et al., 2016*). First, we averaged activity across mismatch trials for each cell. For each mismatch event we then randomly subsampled from control trials (without mismatch events) at identical locations in the tunnel. We repeated this 100 times and subtracted the average trace over repetitions from the real mismatch trace before baseline-correcting for each neuron. Mismatch amplitude was calculated by taking the average of the corrected trace between 0.2 and 1.2 s after mismatch onset. We z-scored each neuron's visual, non-visual, and visuomotor mismatch trace for display purposes only (*Figure 4C*).

Locomotion onset traces were based on running onsets in the non-visual part of the tunnel. Onsets were defined as the first frame in the non-visual section (and after reward delivery) where the average running speed was less than 5 cm/s in a 0.5 s interval preceding the frame, and more than 5 cm/s in the 2 s following the frame. Responses were defined as the baseline-corrected average between 0 and 2 s after onset.

Closed loop visual flow onset traces were made by time-locking (rather than distance) each trial to the start of the visual section. Responses were quantified as the baseline-corrected average between 0.5 and 2 s after trial onset, as the first 20 cm of the tunnel contained no visual stimuli. Only trials that took less than 10 s in the visual section were included in this analysis.

During open loop sessions, the running speed of the mice was uncoupled from visual flow speed, which was fixed at 20 cm/s and followed the regular trial structure. Open loop visual flow onsets were defined as the start of the visual section in trials where the mouse was stationary throughout the visual section. Open loop VF halts were defined as 0.5 s pauses in the VF during stationary trials in open loop condition. For comparison of closed loop mismatch with open loop VF halt we only included trials in closed loop condition during which locomotion speed at mismatch onset was between 5 and 30 cm/s such that the average onset speed was similar as during open loop VF halts (20 cm/s).

Correlations of ChC activity with V-PyCs and NV-PyCs were performed by computing a correlation coefficient on the average $\Delta F/F$ trace of the visual and non-visual section combined between pairs of cells. For each ChC, we then averaged the correlation coefficients of that ChC with all V-PyCs or NV-PyCs in that session.

## Chemogenetic experiments

For passive chemogenetic experiments we used naïve mice injected with AAV9-CaMKIIa-GCaMP6f along with the inhibitory DREADD AAV1-hDlx-dlox-KORD-mCyRFP1 (KORD group) or AAV1-hDlx-dlox-ChrimsonR-tdTomato (control group). We recorded activity of PyCs and ChCs for 15 min under gray screen conditions, after which mice were injected with SalB (10 mg/kg in saline, s.c.). Five minutes later, we again recorded activity for 15 min. Mice were allowed to run or rest freely. For the analysis of overall activity before and after injection of SalB, we calculated the average $\Delta F/F$ of all frames in both recordings for each cell, before averaging over cells for each type. Locomotion modulation curves were computed for each cell by averaging activity over all frames during which the mouse was running at defined speeds within a range. The first bin contained frames during which the mouse was fully stationary (speed =0 cm/s). All subsequent bins contained frames during which the running speed was between that bin and the previous bin (i.e., bin '10' contained speeds 5–10 cm/s). Speeds >20 cm/s were not included in the analysis as several mice did not reach those speeds in either of the sessions. To get the final population curves we averaged over cells for each type. We computed a linear fit over all samples of activity and running speeds for each cell and session and took the coefficients estimates

as the intercept and slope. We then averaged over cells for each type. The LMI was calculated for each cell as follows:

$$LMI = \frac{\Delta F/F_{run} - \Delta F/F_{stat}}{\Delta F/F_{run} + \Delta F/F_{stat}} \tag{13}$$

where $\Delta F/F_{run}$ is the average activity during frames when the mouse was running (>1 cm/s) and $\Delta F/F_{stat}$ is the average activity during frames when the mouse was not running (<1 cm/s). We then computed a correlation coefficient between the LMI and the difference in $\Delta F/F$ due to ChC silencing.

To calculate whether individual neurons were significantly affected by the SalB injection, we compared activity before and after injection for each neuron by comparing the within-session differences in activity (random fluctuations within the pre-session) with the between-session differences in activity (pre versus post) using a permutation test as follows. First, to reduce the correlations between subsequent frames, we averaged activity in 30 frame (~2 s) bins, resulting in 'n' bins. Next, to calculate within-session differences in activity, we split the binned pre-session in two halves and calculated the difference in activity between both halves. We repeated this procedure 'n' number of times, circularly shifting the starting point for splitting the data by one bin on each iteration, resulting in a permutation distribution of n-bin intra-differences in activity. To calculate the between-session differences in activity, we used the same procedure but instead compared the first halves of the pre- and the post-session. In addition, we randomized the starting point for splitting the data of the post-session. We computed significance for each cell using a permutation test on the averaged inter-difference activity values versus the intra-difference permutation distribution.

## In vitro electrophysiology

At ~3 months of age, mice were sacrificed for preparation of acute brain slices containing V1. Mice were deeply anesthetized by application of pentobarbital s.p. (60 mg/kg bodyweight). They were perfused with oxygenated N-methyl-D-glucamine containining artificial cerebrospinal fluid (NMDG-ACSF) containing (in mM) 92 NMDG, 30 NaHCO$_3$, 1.25 NaH$_2$PO$_4$, 2.5 KCl, 20 HEPES, 25 glucose, 0.5 CaCl$_2$, 10 MgCl$_2$ (saturated with 95% O$_2$ and 5% CO$_2$, pH 7.4) and subsequently decapitated. The brain was swiftly removed and submerged in NMDG-ACSF (composition, see above). 300-µm-thick coronal slices were cut using a Leica VT 1200S vibratome (Leica Biosystems, Wetzlar Germany). Slices were allowed to recover for 15 min at 35°C in NMDG-ACSF. They were subsequently transferred to a holding chamber and kept at room temperature in holding ACSF containing (in mM) 125 NaCl, 25 NaHCO$_3$, 1.25 NaH$_2$PO$_4$, 3 KCl, 25 glucose, 1 CaCl$_2$, 6 MgCl$_2$, and 1 kynurenic acid (saturated with 95% O$_2$ and 5% CO$_2$, pH 7.4) until recordings began. Recordings were carried out at ~32°C. Slices were transferred to an upright microscope (BX61WI, Olympus Nederland BV) and constantly perfused with oxygenated ACSF containing (in mM) 125 NaCl, 25 NaHCO$_3$, 1.25 NaH$_2$PO$_4$, 3 KCl, 25 mM glucose, 2 CaCl$_2$, and 1 MgCl$_2$. The chamber was perfused at a rate of 3 ml/min. Cells were visualized with a 40× water immersion objective (Achroplan, NA 0.8, IR 40×/0.80 W, Carl Zeiss Microscopy) with infrared optics and oblique contrast illumination. Patch pipettes were pulled from borosilicate glass (Harvard Apparatus) to an open tip resistance of 4–5 MΩ and filled with intracellular solution containing (in mM) 130 K-gluconate, 10 KCl, 10 HEPES, 4 Mg-ATP, 0.3 Na$_2$-GTP, and 10 Na$_2$-phosphocreatine (pH 7.25, ~280 mOsm) for recordings of ChCs. Patch-clamp recordings were either performed with an Axopatch 200B (Molecular Devices) or a Dagan BVC-700A (Dagan Corporation). Signals were analogue low-pass filtered at 10 kHz (Bessel) and digitally sampled at 50 kHz using an A-D converter (ITC-18, HEKA Elektronik Dr. Schulze GmbH) and the data acquisition software Axograph X (v.1.5.4, Axograph Scientific). Bridge-balance and capacitances were fully compensated in current clamp. Series resistance was compensated to >75% in voltage clamp. The ChCs were identified as mCyRFP-positive neurons in L2/3 of V1 which had a characteristic high-frequency firing pattern (*Figure 1G*). Biocytin (3 mg/ml, Sigma-Aldrich) was routinely added to the intracellular solution to allow for post hoc confirmation of cell morphology and localization (*Figure 1F*).

### Optogenetic stimulation

For mapping of RSC inputs to ChCs, we patched ChCs that were surrounded by a high density of eYFP$^+$ axons. We flashed brief pulses of blue light (10% laser power, resulting in an output intensity of 5.7 mW) with the 470 nm laser line of a laser diode illuminator (LDI-7, 89 North, USA) through the

imaging objective above the cell soma and apical dendrites. We recorded the postsynaptic currents in voltage-clamp mode with a –60 mV holding potential and PSPs were acquired in current clamp mode at *I*=0 holding. For confirmation of monosynaptic inputs (*Cruikshank et al., 2010*), we first washed in TTX to the bath (5 min) to block all AP-induced synaptic release and measured the optogenetically induced PSPs. Next, we applied for 5 min the specific sodium channel blocker TTX (500 nM, Tocris) in combination with potassium channel blocker 4-AP (1 mM, Tocris) to facilitate transmitter release at monosynaptic connections from RSC to ChC and recorded the PSP amplitudes (*Figure 1H*). For characterization of the synaptic connection type (facilitating or depressing), we stimulated with a series of five pulses (5 ms, 20 Hz). We made use of an AAV9-serotyped vector, which has been shown to prevent the artificial synaptic depression during repeated optogenetic stimulation that can occur using other AAV serotypes (*Jackman et al., 2014*).

## Paired recordings

For paired recordings, we first established a ChC recording and then made patch-clamp recording from a nearby PyCs within <50 µm which showed also visually identified bouton cartridges. Pyramidal neurons were filled with high chloride internal solution containing (in mM) 70 K-gluconate, 70 KCl, 0.5 EGTA, 10 HEPES, 4 MgATP, 4 K-phosphocreatine, 0.4 GTP, pH 7.3 adjusted with KOH, 285 mOsmol. Next, we evoked single APs in one cell type with brief current injections (3 ms) and recorded PSPs in the other at their resting membrane potential (*I*=0). Afterward the stimulation of APs was switched to the other neuron to determine reciprocal connectivity. Only responses with 2× the SD of baseline noise (typically 50 µV) were considered being connected.

## Histology and AIS plasticity quantification

For ChC bouton quantification we used five naive and three trained mice injected with AAV9-CaMKII-GCaMP6f and AAV5-hSyn1-flex-ChrimsonR-tdTomato. For AIS quantification, we used an additional two trained mice injected with AAV9-CaMKII-GCaMP6f and AAV1-CAG-flex-mRuby2-GCaMP6f. Mice were perfused with 15 ml of ice-cold PBS followed by 30 ml of 4% paraformaldehyde (PFA). The brains were extracted and post-fixed for 2 hr in 4% PFA at 4°C. Brain were cut in 50 µm slices and selected for staining such that there was ~200 µm between slices. A total of two to three slices containing V1 were used for staining per mouse. We used the following antibodies: Guinea pig-anti-RFP (Synaptic Systems, #390004, 1:500), Rabbit-β-IV-Spectrin (Biotrend, provided by Maren Engelhardt, 1:500), Goat-anti-guinea pig Alexa Fluor 594 (Thermo Fisher, #A11076, 1:1000), Goat-anti-rabbit Alexa Fluor 647 (Thermo Fisher, #A32733, 1:1000). For the staining, slices were blocked for 90 min in 10% normal goat serum (NGS) and 0.5% Triton in PBS followed by incubation in primary antibodies overnight (5% NGS and 0.5% Triton in PBS). After three washing steps, the slices were incubated with secondary antibodies for 120 min at room temperature. After three more washing steps, slices were mounted on SuperFrost Plus glass slides (Fisher Scientific) using fluorescence-preserving mounting medium (Vectashield). Imaging was performed with a Leica SP8 confocal microscope using a 63× N.A. 1.40 oil-immersion objective at a frame size of 1024×1024 pixels. We collected z-stack images at 0.5 µm steps. Z-stacked images were processed in Neurolucida (MBF Bioscience) and AISs and ChC boutons were manually traced under blinded conditions. ChC varicosities were defined as a bouton wherever the axon showed clear thickening alongside the AIS. We only included AISs that had a clear beginning and end.

## Retrograde tracing

Adult Vipr2-Cre mice were unilaterally injected with 90 nl of AAV5-hDlx-DIO-eYFP-t2A-TVA and AAV9-hDlx-DIO-oG (*Lee and Kim, 2019*) in V1 (2.8 mm lateral and 0.5 mm anterior from lambda at a depth of 300 µm) followed by RabV-envA-mCherry 26 days later. After 7 days, mice were perfused as described above and their brains post-fixed overnight in 4% PFA at 4°C. After cryoprotection (48 hr in 30% sucrose at 4°C) and snap freezing in liquid nitrogen slices were stored at –80°C before sectioning. We made 75-µm-thick coronal slices using a cryostat (CM3050 S, Leica) and mounted sections on superFrost Plus glass slides (Fisher Scientific) using Vectashield mounting medium. Imaging for quantification was performed with an Axioscan.Z1 slide scanner (ZEISS, Germany) at 20× and a Leica SP8 confocal microscope using a 40× (NA 1.4) and 63× (NA 1.40) oil-immersion objective at a resolution of 1024×1024.

To quantify the mCherry+ input cells, we followed the QUINT workflow (*Yates et al., 2019*). First, we manually inspected and organized individual images using QuPath and converted them from .czi to .png and .tif in MATLAB. Next, we trained a pixel/object classifier in Ilastik (*Berg et al., 2019*) on a representative subset of images from each mouse and used it to subsequently segment all images for all mice. The output from Ilastik was then recolored using ImageJ. Finally, we registered the slice images to the Allen Brain Reference Atlas using quickNII (*Puchades et al., 2019*) and quantified them using Nutil and MATLAB.

## Statistical analysis

All statistical details for each experiment are shown in figures and figure legends and in *Supplementary file 1*. Statistics on slice physiology experiments were performed using Prism9 (GraphPad Software). All other statistical analyses were performed using LMEM ('fitlme' function in MATLAB) or permutation tests. For LMEM, we considered the response parameter (e.g. visual response magnitude) as a fixed effect and session/mouse as random effect. LMEM takes into account that samples (e.g. neurons) might not be fully independent (e.g. they were obtained from the same session/mouse). We performed a one-way ANOVA on the LMEM followed by a post hoc coefTest using Tukey's HSD to correct for multiple comparisons. Box plots represent the median, the 0.25 and 0.75 quantiles, and the 95% confidence interval of the distributions.

## Acknowledgements

We thank all members of the Levelt lab for discussion and support. We thank staff of the animal facility and mechatronics department at the Netherlands Institute for Neuroscience for technical support. We thank Dr. Seungho Lee from Pohang University of Science and Technology for kindly providing viral vectors for rabies tracing experiments and Dr. Hongkui Zeng and the Allen Institute for Brain Science for providing the Vipr2-Cre mice. We thank Dr. Maren Engelhardt for providing the Rabbit-β-IV-Spectrin antibody, Barbara Hobo for help with tail vein injections, Ulrike Schlegel for help with QUINT software, and David van Oorschot and Dilara Ilhan for help with experiments. We are grateful to Dr. Jean-Charles Paterna and Dr. Melanie Rauch from the Viral Vector Facility VVF at the ETH Zurich and University of Zurich for their expert advice, support, and services regarding viral vector production. This project received funding from the European Union's Horizon 2020 Research and Innovation Program under grant agreement nos. 785907 (HBP SGA2, CL & PR) and 945539 (HBP SGA3, CL & PR), from the Dutch Research Council (NWO-OCENW.KLEIN.178, CL & PR and FlagEra SoundSight, 680-91-320, CL). The authors declare that they have no competing interests. All data needed to evaluate the conclusions in the paper are present in the paper and/or the Supplementary Materials.

## Additional information

### Funding

| Funder | Grant reference number | Author |
|---|---|---|
| European Commission | 785907 | Christiaan N Levelt<br>Pieter R Roelfsema |
| European Commission | 945539 | Christiaan N Levelt<br>Pieter R Roelfsema |
| Nederlandse Organisatie voor Wetenschappelijk Onderzoek | OCENW.KLEIN.178 | Christiaan N Levelt<br>Pieter R Roelfsema |
| Nederlandse Organisatie voor Wetenschappelijk Onderzoek | FlagEra Soundsight 680-91-320 | Christiaan N Levelt<br>Jorrit S Montijn |

The funders had no role in study design, data collection and interpretation, or the decision to submit the work for publication.

## Author contributions
Koen Seignette, Conceptualization, Data curation, Software, Formal analysis, Supervision, Investigation, Visualization, Methodology, Writing – original draft, Project administration, Writing – review and editing; Nora Jamann, Conceptualization, Investigation, Methodology, Writing – original draft, Writing – review and editing; Paolo Papale, Conceptualization, Software, Formal analysis, Investigation, Visualization, Methodology, Writing – original draft; Huub Terra, Investigation, Methodology, Writing – review and editing; Ralph O Porneso, Formal analysis, Investigation, Methodology; Leander de Kraker, Software, Formal analysis, Visualization; Chris van der Togt, Data curation, Software, Formal analysis, Methodology; Maaike van der Aa, Investigation; Paul Neering, Emma Ruimschotel, Investigation, Methodology, Project administration; Pieter R Roelfsema, Supervision, Methodology, Writing – review and editing; Jorrit S Montijn, Formal analysis, Supervision; Matthew W Self, Formal analysis; Maarten HP Kole, Conceptualization, Resources, Software, Visualization, Writing – original draft, Writing – review and editing; Christiaan N Levelt, Conceptualization, Resources, Supervision, Writing – original draft, Project administration, Writing – review and editing

## Author ORCIDs
Koen Seignette (iD) http://orcid.org/0000-0002-7398-6291
Paolo Papale (iD) http://orcid.org/0000-0002-6249-841X
Ralph O Porneso (iD) http://orcid.org/0009-0005-9659-525X
Pieter R Roelfsema (iD) http://orcid.org/0000-0002-1625-0034
Matthew W Self (iD) http://orcid.org/0000-0001-5731-579X
Maarten HP Kole (iD) https://orcid.org/0000-0002-3883-5682
Christiaan N Levelt (iD) https://orcid.org/0000-0002-1813-6243

## Ethics
All experiments were approved by the institutional animal care and use committee of the Royal Netherlands Academy of Arts and Sciences under Central Committee Animal experiments (CCD) licenses AVD 80100 2017 1045, AVD 80100 2022 15934 and AVD 80100 2022 15935.

Reviewer #2 (Public Review): https://doi.org/10.7554/eLife.91153.3.sa1
Reviewer #3 (Public Review): https://doi.org/10.7554/eLife.91153.3.sa2
Author Response https://doi.org/10.7554/eLife.10.7554/eLife.91153.3.3.sa3

# Additional files

## Supplementary files
• Supplementary file 1. Statistics.
• MDAR checklist

## Data availability
Data and scripts to reproduce the figures can be found at https://doi.org/10.17605/OSF.IO/76FK9.

The following dataset was generated:

| Author(s) | Year | Dataset title | Dataset URL | Database and Identifier |
|---|---|---|---|---|
| Seignette K, Jamann N, Levelt C, Kole M | 2023 | Seignette_et_al_eLife_2023 | https://doi.org/10.17605/OSF.IO/76FK9 | Open Science Framework, 10.17605/OSF.IO/76FK9 |

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
