## [Editor Report · eLife assessment]

This **important** work shows **compelling** evidence that Chandelier cells in the visual cortex receive inputs most prominently from local layer 5 pyramidal neurons, only mildly inhibit L2/3 pyramidal neurons, and respond massively to visuomotor mismatch. It also indicates that visual experience in the virtual tunnel activates a plasticity mechanism in Chandelier cells which could be due to the particular visuo-motor coupling experienced in this setting, although a specific control is lacking for this conclusion. This study will be of interest to neuroscientists involved in cortical circuits, visual processing, and predictive coding research.

---

## [Referee Report · Reviewer #2 (Public Review)]

Summary:

Seignette et al. investigated the potential roles of axo-axonic (chandelier) cells (ChCs) in a sensory system, namely visual processing. As introduced by the authors, the axo-axonic cell type has remained (and still is) somehow mysterious in its function. Seignette and colleagues leveraged the development of a transgenic mouse line selective for ChC, and applied a very wide range of techniques: transsynaptic rabies tracing, optogenetic input activation, in vitro electrophysiology, 2-photon recording in vivo, behavior and chemogenetic manipulations, to precisely determine the contribution of ChCs to the primary visual cortex network.

The main findings are (1) the identification of synaptic inputs to ChC, with a majority of local, deep layer principal neurons (PN), (2) the demonstration that ChC is strongly and synchronously activated by visual stimuli with low specificity in naive animals, (3) the recruitment of ChC by arousal/visuomotor mismatch, (4) the induction of functional and structural plasticity at the ChC-PN module, and, (5) the weak disinhibition of PNs induced by ChCs silencing. All these findings are strongly supported by experimental data and thoroughly compared to available evidence.

Strengths:

This article reports an impressive range of very demanding experiments, which were well executed and analyzed, and are presented in a very clear and balanced manner. Moreover, the manuscript is well-written throughout, making it appealing to future readers. It has also been a pleasure to review this article.

In sum, this is an impressive study and an excellent manuscript, that presents no major flaws.

Notably, this study is one of the first studies to report on the activities and potential roles of axo-axonic cells in an active, integrated brain process, beyond locomotion as reported and published in V1. This type of research was much awaited in the fields of interneuron and vision research.

Weaknesses:

There are no fundamental weaknesses; the latter mainly concern the presentation of the main results.

The main weakness may be that the different sections appear somehow disconnected conceptually.

Additionally, some parts deserve a more in-depth clarification/simplification of concepts and analytic methods for scientists outside the subfield of V1 research. Indeed, this paper will be of key interest to researchers of various backgrounds.

---

## [Referee Report · Reviewer #3 (Public Review)]

Summary:

The authors set out to characterize the anatomical connectivity profile and the functional responses of chandelier cells (ChCs) in the mouse primary visual cortex. Using retrograde rabies tracing, optogenetics, and in vitro electrophysiology, they found that the primary source of input to ChCs are local layer 5 pyramidal cells, as well as long-range thalamic and cortical connections. ChCs provided input to local layer 2/3 pyramidal neurons, but did not receive reciprocal connections.

With two-photon calcium imaging recordings during passive viewing of drifting gratings, the authors showed that ChCs exhibit weakly selective visual responses, high correlations within their own population, and strong responses during periods of arousal (assessed by locomotion and pupil size). These results were replicated and extended in experiments with natural images and prediction of receptive field structure using a convolutional neural network.

Furthermore, the authors employed a learned visuomotor task in a virtual corridor to show that ChCs exhibit strong responses to mismatches between visual flow and locomotion, locomotion-related activation (similar to what was shown above), and visually-evoked suppression. They also showed the existence of two clusters of pyramidal neurons with functionally different responses - a cluster with "classically visual" responses and a cluster with locomotion- and mismatch-driven responses (the latter more correlated with ChCs). Comparing naive and trained mice, the authors found that visual responses of ChCs are suppressed following task learning, accompanied by a shortening of the axon initial segment (AIS) of pyramidal cells and an increase in the proportion of AIS contacted by ChCs. However, additional controls would be required to identify which component(s) of the experimental paradigm led to the functional and anatomical changes observed.

Strengths:

The authors bring a comprehensive, state-of-the-art methodology to bear, including rabies tracing, in vivo two-photon calcium imaging, in vitro electrophysiology, optogenetics and chemogenetics, and deep neural networks. Their analyses and statistical tests are sound and for the most part, support their claims. Their results are in line with previous findings and extend them to the primary visual cortex.

Weaknesses:

- Some of the results (e.g. arousal-related responses) are not entirely surprising given that similar results exist in other cortical areas.

---

## [Author Response]

The following is the authors’ response to the original reviews.

We would like the reviewers for their positive and useful comments. Below please ﬁnd our answers to the issues raised.

**Reviewer #1 (Public Review):**
Overall, the experiments are well-designed and the results of the study are exciting. We have one major concern, as well as a few minor comments that are detailed in the following.Major:1. The authors suggest that "Visuomotor experience induces functional and structural plasticity of chandelier cells". One puzzling thing here, however, is that mice constantly experience visuomotor coupling throughout life which is not diﬀerent from experience in the virtual tunnel. Why do the authors think that the coupled experience in the VR induces stronger experience-dependent changes than the coupled experience in the home cage? Could this be a time-dependent eﬀect (e.g. arousal levels could systematically decrease with the number of head-ﬁxed VR sessions)? The control experiment here would be to have a group of mice that experience similar visual ﬂow without coupling between movement and visual ﬂow feedback.Either change would be experience-dependent of course, but having the "visuomotor experience dependent" in the title might be a bit strong given the lack of control for that. We would suggest changing the pitch of the manuscript to one of the conclusions the authors can make cleanly (e.g. Figure 4).

Although the plasticity is induced by the visuomotor experience in the tunnel, we agree that we do not know what aspect of the repeated exposure to the virtual tunnel caused the plasticity. We cannot rule out that it was the exposure to the visual stimuli alone that caused it. Therefore, we rephrased sentences that suggested that it was the coupling between visual stimuli and motor behavior that was responsible for the plasticity. We also changed the title to “Experience Shapes Chandelier Cell Function and Structure in the Visual Cortex”.

We do believe that training the mice in the virtual tunnel does signiﬁcantly increase experience with coupled visuomotor activity, though. In their home cage, mice are mostly active in the dark and there is litle space to run.

Minor:2). "ChCs shape the communication hierarchy of cortical networks providing visual and contextual information." We are not sure what this means.

We thank the reviewer for helping to raise clarity and we rephrased this sentence to:“…ChCs may establish a hierarchical relationship among cortical networks.”

1. "respond to locomotion and visuomotor mismatch, indicating arousal-related activity" This is not clear. We think we understand what the authors mean but would suggest rephrasing.

Agreed. We rephrased this sentence to:"...respond to events that are known to increase arousal levels, such as locomotion and visuomotor mismatch.”

1. 'based on morphological properties revealed that 87% (287/329) of labeled neurons were ChCs" Please specify the morphological properties used for the classiﬁcation somewhere in the methods.

We added that the neurons were positioned at the border of L1 and L2 and had a dendrite reaching into layer 1.

1. We may have missed this - in the patch clamp experiment (Fig.1 H-K), please add information about how many mice/slices these experiments were performed in.

We have added the information to the legend of Fig. 1.

1. "These ﬁndings suggest that the rabies-labeled L1-4 neurons providing monosynaptic input to ChCs are predominantly inhibitory neurons". We are not sure this conclusion is warranted given the sparse set of neurons labelled and the low number of cells recorded in the paired patch experiment. We would suggest properly testing (e.g. stain for GABA on the rabies data) or rephrasing.

We weakened the statement to:“These ﬁndings suggest that the rabies-labeled L1-4 neurons providing monosynaptic input to ChCs may include many inhibitory neurons.”

1. Figure 2E. A direct comparison of dF/F across diﬀerent cell types can be subject to a problematic interpretation. The transfer function from spikes to calcium can be diﬀerent from cell type to cell type. Additionally, the two cell populations have been marked with diﬀerent constructs (despite the fact that it's the same GECI) further reducing the reliability of dF/F comparisons. We would recommend using a diﬀerent representation here that does not rely on a direct comparison of dF/F responses (e.g. like the "response strength" used in Figure 3B). Assuming calcium dynamics are diﬀerent in ChCs and PyCs - this similarity in calcium response is likely a coincidence.

We have removed the quantiﬁcation in this ﬁgure.

1. If ChCs are more strongly driven by locomotion and arousal, then it's a bit counterintuitive that at the beginning of the visual corridor when locomotion speed consistently increases, the activity of ChCs consistently decreases. This does not appear to be driven by suppression by visual stimuli as it is present also in the ﬁrst and last 20cm of the tunnel where there are no visual stimuli. How do the authors explain this?

We do believe that this is suppression driven by visual stimuli. Although on average the strongest visual suppression happens between 20-80 cm, neurons that have their receptive ﬁelds toward the center of the visual ﬁeld will already respond to the stimuli before the mouse reaches the 20 cm location of the tunnel. In addition, although the visual stimuli are the strongest sensory inputs, the background of the visual part of the tunnel has a black and white noise patern, which might already mildly suppress ChC activity. Both arguments are supported by the observation that the visual PyCs (V-PyCs, blue line) in Fig. 4D are already activated at the beginning of the tunnel and that the activity of V-PyCs matches well with the suppression of ChC activity.

1. The authors mention that "ChC responses underwent sensory-evoked plasticity during the repeated visual exposure, even though the visual stimuli were diﬀerent from those encountered during training in the virtual tunnel". How would this work? And would this mean all visual responses are reduced? What is special about the visual experience in the virtual tunnel? It does not inherently diﬀer from visual experience in the home cage, given that the test stimuli (full ﬁeld gratings) are diﬀerent from both.

As mentioned in our answer to point 1, the exposure to visual stimuli is strongly increased since, ﬁrstly, they are presented during the dark phase when the mice are most active and, secondly, they do not get these types of visual inputs in their home cage.

1. Just as a point to consider for future experiments: For the open-loop control experiments, the visual ﬂow is constant (20cm/s) - ideally, this would be a replay of the running speed the mouse previously generated to match statistics.

We agree with this point and will implement replay of earlier sessions in future experiments.

1. We would recommend specifying the parameters used for neuropil correction in the methods section.

This is described on page 24, under “preprocessing”. We also refer to the analysis package (Spectral Segmentation - SpecSeg) in which the neuropil correction as used by us here is explained in more detail.

1. If we understand correctly, the F0 used for the dF/F calculation is diﬀerent from that used for division. Why is this?

We apologize for this mistake, which was based on an older version of the software. We have now corrected this in the revised manuscript.

1. Authors compare neuronal responses using "baseline-corrected average". Please specify the parameters of the baseline correction (i.e. what is used as baseline here).

In the revised version we have now beter explained this in the methods, page 24, under “Passive Sessions”.

**Reviewer #2 (Public Review):**
Summary:Seignete et al. investigated the potential roles of axo-axonic (chandelier) cells (ChCs) in a sensory system, namely visual processing. As introduced by the authors, the axo-axonic cell type has remained (and still is) somehow mysterious in its function. Seignete and colleagues leveraged the development of a transgenic mouse line selective for ChC, and applied a very wide range of techniques: transsynaptic rabies tracing, optogenetic input activation, in vitro electrophysiology, 2-photon recording in vivo, behavior and chemogenetic manipulations, to precisely determine the contribution of ChCs to the primary visual cortex network.The main ﬁndings are 1) the identiﬁcation of synaptic inputs to ChC, with a majority of local, deep layer principal neurons (PN), 2) the demonstration that ChC is strongly and synchronously activated by visual stimuli with low speciﬁcity in naive animals, 3) the recruitment of ChC by arousal/visuomotor mismatch, 4) the induction of functional and structural plasticity at the ChC-PN module, and, 5) the weak disinhibition of PNs induced by ChCs silencing. All these ﬁndings are strongly supported by experimental data and thoroughly compared to available evidence.Strengths:This article reports an impressive range of very demanding experiments, which were well executed and analyzed, and are presented in a very clear and balanced manner. Moreover, the manuscript is well- writen throughout, making it appealing to future readers. It has also been a pleasure to review this article.In sum, this is an impressive study and an excellent manuscript, that presents no major ﬂaws.Notably, this study is one of the ﬁrst studies to report on the activities and potential roles of axo-axonic cells in an active, integrated brain process, beyond locomotion as reported and published in V1. This type of research was much awaited in the ﬁelds of interneuron and vision research.Weaknesses:There are no fundamental weaknesses; the later mainly concern the presentation of the main results. The main weakness may be that the diﬀerent sections appear somehow disconnected conceptually.Additionally, some parts deserve a more in-depth clariﬁcation/simpliﬁcation of concepts and analytic methods for scientists outside the subﬁeld of V1 research. Indeed, this paper will be of key interest to researchers of various backgrounds.
**Reviewer #3 (Public Review):**
Summary:The authors set out to characterize the anatomical connectivity proﬁle and the functional responses of chandelier cells (ChCs) in the mouse primary visual cortex. Using retrograde rabies tracing, optogenetics, and in vitro electrophysiology, they found that the primary source of input to ChCs are local layer 5 pyramidal cells, as well as long-range thalamic and cortical connections. ChCs provided input to local layer 2/3 pyramidal neurons, but did not receive reciprocal connections.With two-photon calcium imaging recordings during passive viewing of drifting gratings, the authors showed that ChCs exhibit weakly selective visual responses, high correlations within their own population, and strong responses during periods of arousal (assessed by locomotion and pupil size). These results were replicated and extended in experiments with natural images and prediction of receptive ﬁeld structure using a convolutional neural network.Furthermore, the authors employed a learned visuomotor task in a virtual corridor to show that ChCs exhibit strong responses to mismatches between visual ﬂow and locomotion, locomotion-related activation (similar to what was shown above), and visually-evoked suppression. They also showed the existence of two clusters of pyramidal neurons with functionally diﬀerent responses - a cluster with "classically visual" responses and a cluster with locomotion- and mismatch-driven responses (the later more correlated with ChCs). Comparing naive and trained mice, the authors found that visual responses of ChCs are suppressed following task learning, accompanied by a shortening of the axon initial segment (AIS) of pyramidal cells and an increase in the proportion of AIS contacted by ChCs. However, additional controls would be required to identify which component(s) of the experimental paradigm led to the functional and anatomical changes observed.Finally, using a chemogenetic inactivation of ChCs, the authors propose weak connectivity to pyramidal cells (due to small eﬀects in pyramidal cell activity). However, these results are not unequivocally supported, as the baseline activity of ChCs before inactivation is considerably lower, suggesting a potentially confounding homeostatic plasticity mechanism might already be operating.Strengths:The authors bring a comprehensive, state-of-the-art methodology to bear, including rabies tracing, in vivo two-photon calcium imaging, in vitro electrophysiology, optogenetics and chemogenetics, and deep neural networks. Their analyses and statistical tests are sound and for the most part, support their claims. Their results are in line with previous ﬁndings and extend them to the primary visual cortex.Weaknesses:Some of the results (e.g. arousal-related responses) are not entirely surprising given that similar results exist in other cortical areas.

We agree that previous studies have shown arousal-related responses of ChC cells and our study conﬁrms those ﬁndings. However, this is not the main message of the article and we present many ﬁndings that are novel.

Control analyses regarding locomotion paterns before and atier learning the task (Figure 5), and additional control experiments to identify whether functional and anatomical changes following task learning were due to learning, repeated visual exposure, exposure to reward, or visuomotor experience would strengthen the claims made.

In ﬁgure 5 we excluded running trials, so locomotion paterns are unlikely to play a major role. We agree that testing what are the factors that contribute to the observed plasticity are important to investigate in future experiments.

The strength of the results of the chemogenetics experiment is impacted by the lower baseline activity of ChCs that express the KORD receptor. At present, it is not possible to exclude the presence of homeostatic plasticity in the network *before* the inactivation takes place.

Although we do not know why there is a diﬀerence in the baseline df/f (e.g. expression levels), we consider it unlikely that expression of the KORD receptor itself without exposure to the ligand causes reduction of ChC activity. Moreover, we are not sure how homeostatic plasticity in the network would occur selectively in KORD-expressing ChCs. Finally, we do not ﬁnd evidence for a relationship between lower ChC calcium signals and the eﬀects of ChC silencing on PyC activity. We performed an additional analysis in which we correlated baseline ChC activity (before salvinorin B injection) with the eﬀect of ChC silencing on PyC activity (post – pre) across mice, and found that this correlation was not signiﬁcant (R = 0.41, p = 0.18).

**Reviewer #1 (Recommendations For The Authors):**
In the spirit of openness of the scientiﬁc discussion, all our feedback and recommendations to the authors are included in the public reviews.
**Reviewer #2 (Recommendations For The Authors):**
Most of my comments and suggestions concern the presentation of the data, to (hopefully) help and convey as clearly as possible the messages of this important article.MainThe main weakness of the paper may be that the diﬀerent sections appear somehow disconnected conceptually. This is particularly true for:-structural plasticity: how can we link this ﬁnding with the rest of the study? Are there ways to correlate this ﬁnding with physiological recordings in individual animals, or to directly test whether particular functional types of PNs (visual, non-visual) undergo plasticity at their AIS?

This is a very interesting question that may be addressed in future experiments.

-the indirect ﬁnding suggesting that ChC weakly inhibits PNs using chemogenetic silencing of PNs. Do chemogenetic manipulations of ChCs aﬀect PN responses in visual paradigm and/or modify the induction of structural plasticity at the ChC-AIS connection?

This is also a very interesting question for future work.

Additionally, some parts would deserve a more in-depth clariﬁcation/simpliﬁcation of concepts and analytic methods (OSI, DSI, MEI...) for scientists outside the subﬁeld of V1 research. Indeed, this paper will be of key interest to researchers of various backgrounds.

In the revised manuscript we brieﬂy explain what an MEI is when ﬁrst introduced, and introduce the abbreviations OSI and DSI at the correct location. We believe orientation and direction selectivity are well-known concepts for the audience reading this article.

MinorThese are discussed by order of appearance in the text.AbstractThe alternative interpretation of error/mismatch negativity to explain ChC activation deserves to appear in the abstract. Arousal consistency in prediction should be in the introduction."In mice running in a virtual tunnel, ChCs respond strongly to locomotion and halting visual ﬂow, suggesting arousal-related activity."This comment holds for the end of the introduction and the beginning of the discussion, as well."These ﬁndings suggest that ChCs provide an arousal-related signal to layer 2/3 pyramidal cells that may modulate their activity". This statement appears to be in contradiction with the weak eﬀect mentioned just before. This comment holds for the end of the introduction.

The full sentence was: “These ﬁndings suggest that ChCs provide an arousal-related signal to layer 2/3 pyramidal cells that may modulate their activity and/or gate plasticity of L2/3 PyCs in V1.” Our results show that activity of layer 2/3 pyramidal cells is modulated (albeit weakly) and it is well possible that ChCs regulate plasticity at the AIS. Therefore, we do not believe that this statement contradicts the weak direct eﬀect of ChCs on layer 2/3 pyramidal cell activity. Therefore , we think that this statement does not contradict the weak direct eﬀect of ChCs on layer 2/3 pyramidal cell activity.

We changed the last sentence of the introduction to “Our ﬁndings suggest that ChCs predominantly respond to arousal related to locomotion or unexpected events/stimuli, and act to weakly modulate activity and/or gate plasticity of L2/3 PyCs in V1.”

Introduction First paragraphComing from a ﬁeld outside of vision research, it is not obvious to me what has been learned from interneuron classes in the past. An example would be welcome in the introduction.

The literature on the role of diﬀerent interneuron types in visual processing and plasticity is too large to pick one or two examples. For the sake of conciseness, we have therefore provided some important references and reviews for the interested readers (references 1 to 10).

Interneuron "subtypes" seem to refer to main classes (e.g. PV+): please rephrase accordingly (ChC being a type and PV+ ChC a subtype).

We changed interneuron “subtypes” to “types” and left L2/3 pyramidal cell “subtypes” unchanged.

Second paragraphBeyond the reversal potential of GABA-ARs at the axon initial segment, GABA may inhibit action potential generation in various conditions (Lipkin et al. 2023, DOI: 10.1523/JNEUROSCI.0605-23.2023 : should be cited).

We added this citation.

Fourth paragraph"ChCs alter the number of synapses at the AIS based on the activity of their postsynaptic targets": the concept of alteration is too vague to let the reader grasp the concept: could the authors rephrase?

We have rephrased the sentence to:

“…ChCs increase the number of synapses at the AIS if their postsynaptic targets are chemogenetically activated…”

Results1. ChCs receive input from long-range sources and L5 PyCs in V1It is not clear how morphological identiﬁcation of ChC was performed. Did dendrites and/or axons of starter cells occasionally overlap as can be expected, complicating the cell-by-cell morphological classiﬁcation?"Most labeled neurons were located on the border between L1 and L2/3 and displayed typical ChC morphology": maybe clarify that this concerns neurons expressing eYFP-TVA?

We assessed the location (at the border of L1 and L2) and spatial distribution of the labeled cells and whether they had a dendrite extending upwards towards into L1. We have now indicated this in the results section and clariﬁed that these neurons express eYFP-TVA.

-Likewise the following would beneﬁt from clariﬁcation " This is further supported by the distributed localization of the labeled neurons": it would also help here to remind the reader of the labelling (presumably retrogradely-labeled mCherrry+ neurons).

We have now clariﬁed in the text that these are mCherry+ neurons labeled by the rabies virus

1. Chandelier cells are modulated by arousal and show high correlations-The authors indicate that the results "(suggest) that ChCs distribute a synchronized signal during high arousal." : it would be stronger to defend this claim by showing a higher ChC-ChC correlation during "arousal" vs. baseline (i.e. analyze high arousal epochs outside of movement). It may be diﬃcult to perform this analysis due to low ﬂuorescence changes outside running episodes, but this should be discussed accordingly. In this respect, the title of the section is more in line with the data presented.

We believe our statement is correct. The activity of ChCs is highly synchronized and their ﬁring rates increase during arousal. We do not state that synchronization increases with arousal.

-A brief explanation of DSI and OSI meaning would be nice for the audience that will deﬁnitely extend beyond vision research given the importance of this study.

See above

1. ChCs are weakly selective to visual information-I may very well miss the point, but the equivalence in response strength among cell classes (Fig3B) seems inconsistent with the wider distribution of high response strength in ChCs (Fig3C). Perhaps a graphical representation taking into account the distribution of single data points in Fig3B would help resolve this discrepancy.

This is because in panel C the response strengths are normalized. We now also state this in the legend to avoid confusion.

-"clearly oriented edge-like paterns with sharp ON and OFF regions": it would help if a representative example was highlighted in Figure 3F.

The majority of L2/3 pyramidal MEIs presented in this panel show this patern.

-It is interesting and surprising that properties of ChCs appear more distinct from those of L5 PNs than from those of L2-3 PNs (Fig 3G-J), given the fact that V1 ChCs were found by the authors to derive their inputs from V1 L5 PNs (please see comments of the discussion for this speciﬁc point).

How ChCs respond based on L5 input depends strongly on how the connections between L5 and ChCs are organized. Similarity between responses of L5 and ChC neurons is not required.

1. Locomotion and visuomotor mismatch drive chandelier cell activity in a virtual tunnel This is the least convincing part in terms of presentation.-It is unclear where/when visuomotor mismatch has been induced in the tunnel: please clarify in the text and in Fig 4B.

We realized that the title of the paragraphs was indeed confusing. In ﬁg. 4A-D and the ﬁrst paragraph about the virtual tunnel, we do not discuss the visuomotor mismatch. This comes later, when we describe the results in Fig. 4E. The titles have been changed.

-No result on visuomotor mismatch is reported in the text of this section, while this is presented in the subsequent section: this needs to be corrected (merge this section with the next?).

We agree, apologies for the confusion. See above.

-It would be interesting to further analyze responses to CS and US. Regarding the US: is water rewarding in non-water-restricted mice? This should be mentioned.

We realized that we did not mention that the mice were water restricted during behavioral training and during the imaging sessions when mice performed the virtual tunnel task. We have now added this to the methods section. Sorry for the omission.

-Along this line: was water sometimes omited? This would provide a complementary way to test the prediction error theory for ChC activation with an alternative modality.

We never omited the water reward. It would be interesting to test this in a future experiment.

1. ChCs have similar response properties as non-visual PyCsIt would help to explicitly mention that in Ai65 mice, only Cre and Flp+ cells express tdTomato (here Vipr2 and PV+).

We added the following sentence:“In these mice, tdTomato was only expressed in cells expressing both Vipr2 and PV.”

1. Visuomotor experience in the virtual tunnel induces plasticity of ChC-AIS connectivityIn relation to the previous section, Jung et al. (doi.org/10.1038/s41593-023-01380-x) recently reported that motor learning reduced ChC-ChC synchrony in M2. Did the author observe a similar change in ChC- ChC synchrony with visual experience/habituation to the task? If available, these data should be reported to help build a clearer picture of ChC functions in the neocortex.

We tested this and also found reduced correlations between ChCs in trained mice vs naïve mice. We added this as text on p14 in the results section.

The low number of ChC boutons' appositions per AIS may be misleading: "While the average number of ChC boutons per AIS remained constant (~2-3 ChC boutons/AIS)"). It would be helpful to make it clear that these are "virally" labelled boutons, as opposed to absolute numbers, if compared with the detailed quantiﬁcation of Schneider-Mizell et al, 2021 (7.4 boutons per AIS in average; doi: 10.7554/eLife.73783.).

We added "virally labeled"

It may be diﬃcult to clearly isolate boutons in light microscopic images of ChC boutons. could the authors comment on this and explain how they solved this issue (in the methods section for instance)?

We elaborated on our deﬁnition of a bouton under confocal microscopy conditions. We also added that the analysis was performed under blinded conditions for the experimenter (i.e. the experimenter did not know whether the images came from trained or untrained mice).

Is there any suggestion for heterogeneity/selectivity for a subset of PNs (the distribution does not seem to show this, though)? It would be interesting to discuss this and try to link this ﬁnding to the rest of the study a bit more directly. Future work could also investigate if genetically deﬁned PN types undergo diﬀerent pre-synaptic plasticity at their AISs (e.g. work cited by the authors by O'Toole et al, 2023 doi: 10.1016/j.neuron.2023.08.015 -this reference can be updated as well, since the work has been published in the meantime).

In our data, we did not ﬁnd evidence for heterogeneity or selectivity of targeting, also not in the physiology using KORD (see below). We do agree that it is an interesting question and deserves atention in future experiments. We also updated the reference.

1. ChCs weakly inhibit PyC activity independent of locomotion speedThe authors state that "recent work in adult mice has reported hyperpolarizing and shunting eﬀects in prelimbic cortex, S1 and hippocampus (18, 26, 27)": however, to my knowledge studies presented in refs 26 & 27 found reduced activity/ﬁring of PNs upon optogenetic activation of ChCs in vivo, but did not perform intracellular recordings to assess GABA-A reversal potential at the AIS. I would like to kindly ask the authors to correct this sentence.If the polarity of responses is discussed, they may rather refer to the corresponding literature including Rinetti Vargas et al (doi: 10.1016/j.celrep.2017.06.030), Lipkin et al (doi: 10.1523/JNEUROSCI.0605- 23.2023), and Khirug et al (doi: 10.1523/JNEUROSCI.0908-08.2008.).

We added the reference to Lipkin et al and changed the sentence so that it matches the references..

In an atempt to link ﬁndings from several parts of the article, did the authors investigate whether chemogenetic eﬀects were diﬀerent in visual vs non-visual PNs? As ChCs are functionally related to visual PNs, one might indeed speculate that these cells are synaptically connected.

We did not ﬁnd evidence for selectivity in the chemogenetic eﬀect. We compared the chemogenetic eﬀect to locomotion modulation (see text accompanying Fig 7.) – based on our observation that non- visual PyCs were more strongly modulated by locomotion (see Fig. 4) – but did not ﬁnd any signiﬁcant correlation.

" We ﬁrst looked at the average activity of neurons in both essions.": sessions

Thank you for noticing. We corrected this.

DiscussionSummary of ﬁndings-It would be worthwhile to include in the summary the ﬁnding of mismatch-related activity, that appears to explain more convincingly ChC activation than arousal per se (with the data available).

We updated the summary of the discussion accordingly.

-Moreover, the last part of the article (weak inhibition of PNs by ChCs), despite being very important, is not mentioned.

We now mention this in the summary of the discussion (“Finally, ChCs only weakly inhibit PyCs.”)

Discussion of ﬁndings-" Optogenetic activation of cortical feedback": it is not clear what the authors mean by cortical feedback. As RS was retrogradely labeled, this region may rather provide feedforward inhibition to V1 via ChCs.

Retrosplenial cortex is a higher order cortical area and only provides feedback to V1.

-"This means that each ChC receives input from many L5 PyCs, which could explain the low selectivity of ChC responses we observed to natural images compared to those of L2/3 and L5 PyCs". : perhaps state explicitly that the convergence of many PN inputs each carrying diﬀerent RF/visual properties "averages out" in ChC (as you do a few lines below for MEI).

At this point, we do not know how the connections from L5 to ChCs are organized. Whether this converge results in “average out” is therefore not so certain. We have made an atempt to clarify the situation. (“This convergence of L5 PyC inputs, if not strongly organized, could explain the low selectivity of ChC responses we observed to natural images compared to those of L2/3 and L5 PyCs.”)

-"However, we did not identify neuromodulatory inputs to ChCs in our rabies tracing experiment. Possibly, these inputs act predominantly through extrasynaptic receptors and were therefore not labeled by the transsynaptic rabies approach.": here, the authors shouldcite the work by Lu et al (doi: 10.1038/nn.4624) which found basal forebrain (diagonal band of Broca) cholinergic inputs to ChC of the PFC in the Nkx2.1CreER mouse model. Moreover, the authors should discuss potential technical diﬀerences (?) responsible for this discrepancy. Beyond the extrasynaptic release of neuromodulators, rabies strains may display diﬀerent tropism proﬁles for neuron classes.

We have now added a sentence discussing this and added the reference in the revised manuscript.

-The section dedicated to prediction error is particularly interesting and relevant. In my opinion, this interpretation should be further emphasized in the abstract and summary of ﬁndings paragraph in the discussion (as already indicated).

Yes, we agree and have added some emphasis.

-" These ﬁndings are thus in contrast with the general notion that ChCs exert powerful control over PyC output (28, 78), but consistent with computational simulations predicting a relatively small inhibitory eﬀect of GABAergic innervation of the AIS, possibly involving shunting inhibition (79, 80)."These ﬁndings are also consistent with results from PFC and dCA1 studies showing, with electrophysiological recordings combined with optogenetic stimulation of ChCs, that a small proportion of putative PNs was inhibited upon ChC stimulation (doi: 10.1038/nn.4624 doi: 10.1016/j.neuron.2021.09.033).Perhaps the eﬀect of ChCs is limited in all these experiments by a suboptimal eﬃciency of ChC targeting. Moreover, inhibition might be restricted to a subset of PNs carrying a speciﬁc function. This could be discussed.

We added an explanation for the weak eﬀects of silencing to the discussion and stated that our results are in line with ﬁndings in PFC and CA1. (“One explanation for the weak eﬀects we observed is the high variability in the number of GABAergic boutons that PyCs receive at their AISs. Possibly, only a smaller fraction of PyCs with high numbers of AIS synapses are inhibited when ChCs are active. Indeed, we ﬁnd that only a small fraction of PyCs increased their activity upon chemogenetic silencing of ChCs, in line with ﬁndings by others showing that manipulating ChC activity in vivo has relatively weak eﬀects on small populations of PyCs (27, 28).”)

Although we cannot rule out that ChC targeting is suboptimal in our and other experiments, the expression of the KORD receptor as visualized by mCyRFP1 ﬂuorescence appeared very strong. In addition, the common notion in the ChC ﬁeld is that ChCs exert powerful control over PyC ﬁring. Even suboptimal labeling should in that case show clear inhibitory eﬀects. Similar experiments with PV+ interneurons would show very convincing inhibition, even if labeling is suboptimal. To keep the discussion concise, we prefer to leave this particular point out.

-" ChC activation could prevent homeostatic AIS shortening of L2/3 PyCs if their activity occurs during behaviorally relevant, arousal inducing events": this postulate seems to be very interesting but is not very clear and lacks some mechanistic speculation.

We considered elaborating more on this hypothesis. However – given that it is merely a speculation at this point – we do not wish to lengthen the discussion further on this point.

A reference to previous studies demonstrating high levels of synchronous ChC activities is missing: the authors may cite Dudok et al., Schneider-Mizell et al., and Jung et al. (and discuss a change in synchrony with learning or habituation in the case of this study; see above).

We have now also referred to these papers in the context of high correlations between ChCs.

MethodsBeyond references to reagents (eg antibodies, viruses), lot numbers should be provided whenever this is possible. Indeed, there might be strong lot-to-lot variations in speciﬁcity and eﬃciency.
**Reviewer #3 (Recommendations For The Authors):**
Major:(Figure 5) Control analysis missing. Mice before and after training in VR will almost deﬁnitely exhibit diﬀerent running paterns when viewing driftng gratings. Since ChCs are strongly modulated by locomotion, assess whether results depend on changes in running.

Although we did not compare locomotion paterns before and after training, we removed all trials in which the mice were running (see methods). Therefore, we can exclude that these results are caused by changes in running behavior.

(Figure 5 & 6) What would happen with simple passive visual experience, not in a visuomotor task? What if there was no reward? What if there was an open-loop experiment with random reward? To which speciﬁc aspect of the experiment are the results atributable?

These are indeed very interesting questions that may be tested in future experiments.

(Figure 7 B, H) The pre-injection ChC activity in the KORD group is less than 50% of that in control mice! Discuss the eﬀect of such a shift in baseline. Plasticity of PyCs even before ChC inactivation?

See answer to the above question in the public section of reviewer 3.

(Figure 3 H) Contrast tuning results, as far as I understand, come only from the CNN. However, if I understood correctly, during the passive viewing of gratings there were already diﬀerent contrasts. Why not show contrast tuning there? Do the results disagree?

We did indeed show stimuli at diﬀerent contrasts during the passive viewing of gratings. Although the results from those recordings were not optimal for deﬁning contrast sensitivity, they also showed that ChC responses were less modulated by contrast than PyCs.

Minor:(Figure 3) Explain the potential impact of diﬀerent indicators 8m vs 6f due to diﬀerent baselines and dynamics.

We believe there is no impact of diﬀerent indicators, because for the CNN analyses we estimated spikes using CASCADE. This toolbox is speciﬁcally designed to generalize across diﬀerent calcium indicators.Although GCaMP8m was not included in their training set, the wide variety of indicators used provides a solid basis for generalizable spike estimation.Importantly, comparisons between L2/3 PyCs and ChCs also would not be aﬀected by this concern.

(Figure 4) NV-PyCs. Would you call all of these mismatch-responsive neurons? Discuss the diﬀerence in the percentage of neurons (more than 50% of total PyCs here, compared to signiﬁcantly less - up to 40% in previous studies, as far as I'm aware)

Not all NV-PyCs appeared to be mismatch-responsive neurons.

(Figure 6 D) No error bars?

This is a representation of the fraction of all contacted AISs, which has no error bars indeed.

(Figure 6 E-F and H-I) These pairs of panels contain essentially the same information. The ﬁrst panel of each pair seems redundant.

We prefer to keep both plots in place, as in this case the skewness of the histogram can be helpful, which is less clear in the boxplot (which in itself displays the quantiles beter).

The equation for direction tuning still has ang_ori, instead of ang_dir which I'm assuming should be there.

Thank you for noticing, we corrected it.

The response for drifting gratings is calculated from a diﬀerent interval (0.2-1.2s) compared to natural images (0-0.5s). Why?

Because we used spike probability in the case of the natural images to shorten the signal, and the visual stimuli were presented for 0.5 s (instead of 1 s as with the gratings).

Very minor:It would be helpful for equations to have numbers.

Done

Sparsity equation. Beter to have it as a general equation, with N instead of 40. Then below it can be explained that N is the number of images = 40.

Done

"The similarity of these MEIs with those we found for ChCs is in line with the idea that ChCs are driven by input from a large number of L5 PyCs (but do not exclude alternative explanations)." - in parenthesis it should be *does* not exclude.

Corrected.

"In contrast, the response strength of PyCs was only mildly and non-signiﬁcantly reduced after training"*statistically* non-signiﬁcant..

Corrected.

"We ﬁrst looked at the average activity of neurons in both essions." - *sessions*

Corrected.

(Figure 7 C) Explain what points and error bars represent

Done.